# Global atmospheric carbon monoxide budget 2000–2017 inferred from multi-species atmospheric inversions

Bo Zheng[1], Frederic Chevallier[1], Yi Yin[2], Philippe Ciais[1], Audrey Fortems-Cheiney[1], Merritt N. Deeter[3], Robert J. Parker[4,5], Yilong Wang[1], Helen M. Worden[3], and Yuanhong Zhao[1]

[1]Laboratoire des Sciences du Climat et de l'Environnement, CEA-CNRS-UVSQ, UMR8212, Gif-sur-Yvette, France
[2]Division of Geological and Planetary Sciences, California Institute of Technology, Pasadena, CA, USA
[3]Atmospheric Chemistry Observations and Modeling Laboratory, National Center for Atmospheric Research, Boulder, CO, USA
[4]Earth Observation Science, Department of Physics and Astronomy, University of Leicester, Leicester, UK
[5]National Centre for Earth Observation, University of Leicester, Leicester, UK

*Correspondence to*: Bo Zheng (bo.zheng@lsce.ipsl.fr)

**Abstract.** Atmospheric carbon monoxide (CO) concentrations have been decreasing since 2000 as observed by both satellite- and ground-based instruments, but global bottom-up emission inventories estimate increasing anthropogenic CO emissions concurrently. In this study, we use a multi-species atmospheric Bayesian inversion approach to attribute satellite-observed atmospheric CO variations to its sources and sinks in order to achieve a full closure of the global CO budget during 2000–2017. Our observation constraints include satellite retrievals of the total column mole fraction of CO, formaldehyde (HCHO), and methane ($CH_4$) that are all major components of the atmospheric CO cycle. Three inversions (i.e., 2000–2017, 2005–2017, and 2010–2017) are performed to use the observation data to the maximum extent possible as they become available and assess the consistency of inversion results to the assimilation of more trace gas species. We identify a declining trend in the global CO budget since 2000 (three inversions are broadly consistent during overlapping periods), driven by reduced anthropogenic emissions in the U.S. and Europe (both likely from the transport sector), and in China (likely from industry and residential sectors), as well as by reduced biomass burning emissions globally, especially in Equatorial Africa (associated with reduced burned areas). We show that the trends and drivers of the inversion-based CO budget are not affected by the inter-annual variation assumed for prior CO fluxes. All three inversions estimate that surface CO emissions contradict the global bottom-up inventories in the world's top two emitters for the sign of anthropogenic emission trends in China (e.g., here $-0.8 \pm 0.5\%$ $yr^{-1}$ since 2000 while the prior gives $1.3 \pm 0.4\%$ $yr^{-1}$) and for the rate of anthropogenic emission increase in South Asia (e.g., here $1.0 \pm 0.6\%$ $yr^{-1}$ since 2000 smaller than $3.5 \pm 0.4\%$ $yr^{-1}$ in the prior inventory). The posterior model CO concentrations and trends agree well with independent ground-based observations and correct the prior model bias. The comparison of the three inversions with different observation constraints further suggests that the most complete constrained inversion that assimilates CO, HCHO, and $CH_4$ has a good representation of the global CO budget, therefore matches best with independent observations, while the inversion only assimilating CO tends to underestimate both the decrease in anthropogenic CO emissions and the increase in the CO chemical production. The global CO budget data from all three inversions in this study can be accessed from https://doi.org/10.6084/m9.figshare.c.4454453.v1 (Zheng et al., 2019).

# 1 Introduction

Carbon monoxide (CO) is present in trace quantities in the atmosphere but plays a vital role in atmospheric chemistry. CO is part of a photochemical reaction sequence driven by hydroxyl radical (OH) that links methane ($CH_4$), formaldehyde (HCHO), ozone ($O_3$), and carbon dioxide ($CO_2$). The reaction of CO with OH accounts for 40% of the removal of OH in the troposphere (Lelieveld et al., 2016) and governs the oxidizing capacity of the Earth's atmosphere. This reaction also makes CO an important precursor of $O_3$ and affects the $CH_4$ lifetime and abundance, which leads to an indirect positive radiative forcing of 0.2 W m$^{-2}$ (Myhre et al., 2013). Since CO is heavily involved in the relationship between atmospheric chemistry and climate forcing, it is crucial to investigate its atmospheric burden, trends, and the underlying drivers.

Atmospheric CO concentrations following industrialization increased until around the late 1970s and have been decreasing since the 1990s as shown by Greenland firn air records (Wang et al., 2012; Petrenko et al., 2013) and surface flask samples collected at few sites (Khalil and Rasmussen, 1994; Novelli et al., 2003; Gratz et al., 2015; Schultz et al., 2015). These few measurement points may not capture the global trend, given the short lifetime of CO being of only several weeks. Since 2000, the atmospheric CO burden has been monitored globally and continuously by the CO vertical profiles and tropospheric total columns retrieved from the space-borne Measurements Of Pollution In The Troposphere instrument (MOPITT, Deeter et al., 2017). A declining trend is apparent in the MOPITT data, more pronounced in the Northern Hemisphere (Worden et al., 2013) where most of the global economic activity occurs associated with a large amount of fossil fuel use and CO emissions from combustion processes. An intuitive explanation of the declining CO is that the improvement of combustion technologies (e.g., high-efficiency engines) has reduced CO emissions over time, but global bottom-up inventories oppositely estimate increasing anthropogenic CO emissions after 2000 because of the increasing fossil fuel consumption (Granier et al., 2011; Crippa et al., 2018; Hoesly et al., 2018). When prescribed with these inventories, atmospheric chemistry models fail to capture the observed rapid decline in atmospheric CO burdens globally (Petrenko et al., 2013; Strode et al., 2016).

Interpreting atmospheric CO trends requires accurate quantification of the global CO budget (Duncan et al., 2007), including surface sources, atmospheric sources (oxidation of hydrocarbons, known as CO chemical production), and atmospheric sinks. The surface sources include anthropogenic incomplete combustion of fossil fuels and biofuels (Hoesly et al., 2018), biomass burning (van der Werf et al., 2017), plant leaves (Tarr et al., 1995; Bruhn et al., 2013), and the ocean (Conte et al., 2019). Anthropogenic emissions depend on fuel type, fuel amount, combustion technology, and emission control devices (e.g., a catalytic converter for an automobile). Biomass burning emissions are caused by human-igniting or lightning fires on fire-prone landscapes such as savannas and forests. Fire intensities and CO emissions are sensitive to climatic conditions such as drought and heat wave (Chen et al., 2017), and are enhanced (e.g., deforestation) or suppressed (e.g., cultivation or forest fire suppression) by human activities. Peat fires produce larger CO emissions from incomplete combustion than open fires, especially in Indonesia (Yin et al., 2016). A small amount of CO is directly generated from plant leaves and marine biogeochemical cycling, which vary less from year to year than anthropogenic and biomass burning sources. However, large amounts of CO are produced from the oxidation of hydrocarbons from biogenic emissions that can vary due to climate and

human land use changes. The oxidation by OH is the dominant sink of CO and gives CO a global average chemical lifetime of 1–3 months (Seinfeld and Pandis, 2006).

The contradiction between growing anthropogenic CO emissions in global bottom-up inventories (Granier et al., 2011; Crippa et al., 2018; Hoesly et al., 2018) and the MOPITT-observed declining CO since 2000 suggests three scenarios: 1) the CO sink has been increasing faster than the CO source, and 2) the bottom-up inventories underestimate the improvement in combustion technology and the actual decrease of anthropogenic CO emissions, and 3) biomass burning emissions or CO chemical production have been decreasing rapidly. The first scenario is unlikely to be the main reason, because OH is considered well buffered in the atmosphere with a small inter-annual variation (Montzka et al., 2011; Naik et al., 2013; Voulgarakis et al., 2013), or slightly decreasing in the last decade, a process that may partly explain the renewed growth of atmospheric $CH_4$ since 2007 (Rigby et al., 2017; Turner et al., 2017; McNorton et al., 2018). It is difficult to simply determine whether scenarios 2) and 3) play a big or small role because the estimates of anthropogenic and biomass burning CO emissions and trends are typically subject to large uncertainties. Although at first sight, growing anthropogenic emissions seemingly disagree with the observed declining CO, trends in biomass burning emissions, CO chemical production, and atmospheric transport all play a confounding role. This calls for a full closure of the atmospheric CO budget using the best available data and knowledge, which can be framed as an inverse problem that matches all available information within their uncertainties.

The main purpose of this study is to reconcile the observed and bottom-up estimated atmospheric CO budget since 2000, and to provide a self-consistent and accurate inversion-based data product of the global CO budget during 2000–2017. We use an atmospheric Bayesian inversion approach to infer the global CO budget, where surface CO emissions, CO chemical production, and CO sinks are optimized at a spatial resolution of 3.75° longitude × 1.9° latitude every 8 days. Three inversions (2000–2017, 2005–2017, and 2010–2017, see details in Sect. 2.2) are performed assimilating multiple satellite observations of CO, HCHO, and $CH_4$ in the inversion system as they become available in order to constrain the CO reaction sequence. One additional sensitivity inversion is conducted to use flat prior CO fluxes without inter-annual variations in order to assess the influence of prior variations on the CO budget estimates. Based on these inversion results, we investigate the magnitudes, trends, and drivers of the global CO budget from 2000 to 2017, helping to understand the observed remarkable decline in the atmospheric CO since 2000.

## 2 Methods

### 2.1 General methodology

The evolution of atmospheric CO concentrations with time ($\partial[CO]/\partial t$) in a 3-D atmospheric model grid box is expressed as the sum of multiple CO emission sources ($Source_{CO}$) minus the CO sink ($Sink_{CO}$), which can be represented by the following equation (1).

$$\frac{\partial [CO]}{\partial t} = \sum \left( Source_{CO} \right) - Sink_{CO}$$

$$= -\mathbf{v} \bullet \nabla [CO] + \sum_{sector} \left( E_{CO} \right) + P_{CH_4 \to CO} + P_{NMVOCs \to CO} - k_{CO+OH}(T)[CO][OH] - Dep_{CO} \tag{1}$$

The flux divergence term ($\mathbf{v} \bullet \nabla [CO]$) represents the transport of CO into and out of each atmospheric model grid box, whose sum is equal to zero at the whole globe. $E_{CO}$ is the surface CO emission flux from different source sectors (i.e., anthropogenic, biomass burning, biogenic, and oceanic). $P_{CH4 \to CO}$ and $P_{NMVOCs \to CO}$ represent the CO chemical production from $CH_4$ and non-

methane volatile organic compounds (NMVOCs), respectively, oxidized by OH in the atmosphere. The CO chemical sink ($k_{CO+OH}(T)[CO][OH]$) is calculated on the basis of CO ($[CO]$), OH ($[OH]$), and a temperature ($T$)-dependent rate ($k_{CO+OH}$), and $Dep_{CO}$ is the dry deposition of CO that contributes about 7% of the CO total sink (Stein et al., 2014).

We use the global 3-D transport model of the Laboratoire de Météorologie Dynamique (LMDz) coupled with a simplified chemistry module, Simplified Atmospheric Chemistry assimilation System (SACS) (Pison et al., 2009), to simulate the

atmospheric physical and chemical processes described in Eq. (1) except the dry deposition not represented by this model. An atmospheric Bayesian inversion framework is built upon the LMDz-SACS model (Chevallier et al., 2005, 2009), and satellite observations of the relevant trace gas species (CO, HCHO, and $CH_4$) are assimilated to constrain the inversion system (Zheng et al., 2018a, 2018b) given some prior information on the initial model state, surface emissions, CO chemical production, and OH field. Section 2.2 provides details of the atmospheric inversion approach and the model evaluation

protocol, and Sect. 2.3 describes how we analyse the global CO budget using inversion results.

## 2.2 Atmospheric Bayesian inversion

The core of atmospheric Bayesian inversion is the minimization of the following cost function:

$$J(\mathbf{x}) = \left( \mathbf{x} - \mathbf{x^b} \right)^T \mathbf{B}^{-1} \left( \mathbf{x} - \mathbf{x^b} \right) + \left( H(\mathbf{x}) - \mathbf{y} \right)^T \mathbf{R}^{-1} \left( H(\mathbf{x}) - \mathbf{y} \right) \tag{2}$$

$\mathbf{x}$ is the control vector that gathers the target variables we seek to optimize, and $\mathbf{x^b}$ is a prior guess of these variables assuming

unbiased Gaussian error statistics represented by a covariance matrix $\mathbf{B}$. $\mathbf{y}$ is the observation vector containing all the observation data assimilated to constrain the inverse problem; their error statistics are assumed to be unbiased and Gaussian with a covariance matrix $\mathbf{R}$. $H$ is the forward model (the combination of the LMDz-SACS model, a sampling operator, and an averaging kernel operator) that calculates the equivalent of the observation data in $\mathbf{y}$ based on the control vector $\mathbf{x}$. The forward model error and the representation error caused by the mismatch between model and observation resolutions are also included

in $\mathbf{R}$, making $\mathbf{R}$ represent a combination of measurement, forward model, and representation errors. Configurations of the variables and vectors in Eq. (2) are summarized in Table 1 and Table S1, most of which have already been described in our previous papers (Zheng et al., 2018a, 2018b). To solve the inverse problem, forward and adjoint codes are iteratively run until sufficient convergence of the cost function (Eq. (2)), and the last iteration with optimized model states gives us the best estimate that matches all available information within their uncertainties.

The inversion system has several updates compared to the version developed by the same research team a few years ago to study atmospheric CO trends (Yin et al., 2015). First, we use a higher resolution transport model with finer horizontal grid cells ($3.75° \times 1.9°$ compared to $3.75° \times 2.5°$) and more vertical layers (39 layers compared to 19 layers) than Yin et al. (2015) used. Second, we assimilate the MOPITT version 7 data as an observation constraint, which is improved with respect to

retrieval biases and bias drift compared to the previously used version 6 data (Deeter et al., 2017). Third, we use the latest global bottom-up emission inventories as prior, including the Community Emissions Data System (CEDS, Hoesly et al., 2018) for the anthropogenic source and the Global Fire Emissions Database (GFED) 4.1s for the biomass burning source. The regional studies of Zheng et al. (2018a, 2018b) used the same configuration than here, except that they assimilated in-situ measurement for $CH_4$ rather than satellite retrievals.

We perform four inversion simulations with our inversion system (Table 2). Inversion #1 (2000–2017) is constrained by CO total columns derived from the MOPITT version 7 TIR-NIR retrievals; Inversion #2 (2005–2017) is constrained by both MOPITT CO column and Ozone Monitoring Instrument (OMI) version 3 HCHO column; and Inversion #3 (2010–2017) is additionally constrained by Greenhouse gases Observing SATellite (GOSAT) column-averaged dry air mole fractions of $CH_4$ ($XCH_4$). All three inversions also assimilate in-situ measurement of methyl chloroform (MCF) to help constrain OH (Yin et

al., 2015), but the rapidly declining levels of MCF in the atmosphere makes this constraint progressively ineffective (e.g., Liang et al., 2017). Factorial simulations constrained by MOPITT CO, OMI HCHO, and GOSAT $XCH_4$ allow us not only to better constrain the photochemical reaction sequence of CO, but also to facilitate a quantitative assessment of potential uncertainties in the inversion CO budget relating to the use of different observation constraints. We also do a sensitivity Inversion #4 (2000–2017) with the same observation constraints as Inversion #1 but flat prior surface CO emissions without

any inter-annual variability to check if the derived CO budget is robust to the inter-annual variation of the prior CO fluxes. Inversion results of the optimized CO concentration in the atmosphere are evaluated against independent ground-based observations from the World Data Centre for Greenhouse Gases (WDCGG, https://gaw.kishou.go.jp/) and the Total Carbon Column Observing Network (TCCON, Wunch et al., 2011). The WDCGG provides measurements of surface CO concentrations through in-situ and flask sample measurements, and the TCCON provides retrievals of the column-averaged

dry air mole fraction of CO (XCO). We collect observation data from 110 sites in WDCGG (Table S2) and from 32 sites in TCCON (Table S3, station names and references are shown in Fig. A1), which cover the whole globe (Fig. A1). To do the evaluation, we first sample the model at the location and time of the observation data, and then calculate the average values and annual trends for both model and observation. The annual trends are estimated on the basis of monthly time series using a curve fitting method (https://www.esrl.noaa.gov/gmd/ccgg/mbl/crvfit/crvfit.html), which is also used in Zheng et al. (2018a).

$p$ values and 95% confidence intervals are given to assess the robustness of the estimated trends. The metrics used for evaluation include normalized mean bias (NMB), root mean square error (RMSE), the Pearson's correlation (R), and the regression slope between model and observation among all surface sites.

## 2.3 Atmospheric CO budget

The picture of atmospheric CO budget derived from our inversions includes surface fluxes (the sum of direct emissions from different source sectors and of dry deposition), CO chemical production, and CO chemical sink. Given the marginal role played by dry deposition (about 20% of the direct emissions, Stein et al. 2014), the inferred surface fluxes will be assumed to be made of direct emissions only in the following. The CO chemical production and chemical sink are direct outputs from the inverse system, which calculates the CO yield from the oxidation of $CH_4$ and of NMVOCs and the CO oxidation sink with the linearized chemistry scheme in each model grid box at each time step of the model simulation.

To obtain sectoral surface emission fluxes, we multiply the optimized 8-daily surface total fluxes by the proportion of each sector in each model grid cell as given by the prior (Jiang et al., 2017; Yin et al., 2016; Zheng et al., 2018b). We distinguish between four source sectors: anthropogenic, biomass burning, biogenic, and oceanic, which have rather distinct spatial-seasonal patterns in CO emission distributions. Without considering the oceanic emission, 96% of the CO emissions on land are distributed on grid cells with a dominant emission source (i.e. a sector that contributes more than 50% of CO flux in that grid). Further, 85% of CO emissions on land are distributed in grid cells where such a dominant source accounts for more than 65% of the CO total flux. The distinct seasonal evolutions of different sectors also help attribute emissions to one specific source sector. For example, the biomass burning in Africa typically accounts for 80–90% of surface CO emissions in the dry season (Zheng et al., 2018b). This local source homogeneity reduces the attribution bias of sectoral CO fluxes, although it cannot eliminate all biases. To minimize remaining biases, we focus on annual emission anomalies through removing multiannual average or calculating linear trends to reduce the systematic errors. This makes it possible to directly compare the inversion-based emissions with bottom-up inventories to analyze the underlying emission drivers (Sect. 4.2).

We also collect previous top-down inversion estimates of the global CO budget from the scientific literature (Table S4). These studies used older versions of MOPITT retrievals (or other satellite data) and coarser-resolution transport models, and they did not have the capability of multi-species constraints. Despite different observation data quality and inversion model set up, it is still meaningful to do such a review to visualize the evolution of the inversion-based global atmospheric CO budget.

## 3 Results

### 3.1 Observed declining CO since 2000

Tropospheric CO columns observed by MOPITT v7 declined at an average rate of $-0.32\pm0.05\%$ $yr^{-1}$ ($p<0.01$; numbers being ± signs are 95% confidence limits from a linear fit) from 2000 to 2017 over the whole globe (Fig. 1a; Fig. S1). This trend, equivalent to $-5.6\times10^{15} \pm 0.9\times10^{15}$ mol $cm^{-2}$ $yr^{-1}$, is much larger than the retrieval bias drift of $1.0\times10^{15} \pm 1.0\times10^{15}$ mol $cm^{-2}$ $yr^{-1}$ in the MOPITT v7 TIR-NIR product (Deeter et al., 2017), indicating that the observed downward trend is robust to satellite retrieval errors (Worden et al., 2013). The distributed ground-based sites of the WDCGG network (dots in Fig. 1a) also measure rapidly decreasing surface CO concentrations during 2000–2017, broadly consistent with the negative trends of the MOPITT

CO columns. The sites located on or near continents, more affected by land sources upwind, generally show faster CO concentration declines than the MOPITT CO columns, while the background sites, especially those over islands, agree better with MOPITT observations. The TCCON observed XCO also presents consistent declining trends with MOPITT (Fig. S2).

The trends in MOPITT CO columns reveal heterogeneous spatial patterns (Figs. 1b and 1c). The largest decrease is seen in the northern mid-latitudes (30° N–60° N), where Canada (CAN), USA, Europe (EU), Russia (RUS), and China (CHN) all present statistically significant declining trends (Fig. 1c). The tropical region (30° S–30° N) has a smaller decrease in CO columns due to some increasing trends existing over South Asia (SAS) and over a large part of Africa. South Asia is the only region that has a statistically significant trend of rising CO columns since 2000 according to our region splitting (Fig. A1). The African continent presents both increasing and decreasing CO columns that compensate each other and therefore lead to no statistically significant trends over Equatorial Africa (EQAF) and Southern Africa (SAF) (Fig. 1c). This is consistent with ground-based observations at Ascension Island, UK (ASC, Table S2) located downwind of the West African coast that sees CO plumes from Africa. The southern mid-latitudes (30° S–60° S) primarily consist of the ocean with very few lands, where the MOPITT and WDCGG observations both show consistently moderately decreased CO abundance.

## 3.2 Global atmospheric CO budget

### 3.2.1 MOPITT constrained inversion

Inversion #1, constrained by MOPITT v7 CO columns, estimates that the global annual CO source was $2.6 \times 10^3$ Tg CO yr$^{-1}$ on average and decreased at a rate of $-10.0 \pm 6.9$ Tg CO yr$^{-2}$ ($p < 0.01$) during 2000–2017 (Table 3). The chemical sink of CO by reaction with OH is estimated as $2.6 \times 10^3$ Tg CO yr$^{-1}$ with a declining trend of $-9.3 \pm 6.0$ Tg CO yr$^{-2}$ ($p < 0.01$). The steady declining CO source breaks the source-sink balance of CO in the atmosphere, makes the CO source slightly smaller than its sink, and therefore drives the atmospheric CO burden down from 2000 to 2017. The reduced CO source further leads to a comparable decline in the CO sink due to relatively stable OH burden in the troposphere.

The global CO source is spread roughly equally between direct emissions from the Earth's surface ($1.4 \times 10^3$ Tg CO yr$^{-1}$) and the chemical production in the atmosphere ($1.2 \times 10^3$ Tg CO yr$^{-1}$). The direct emissions are estimated to have decreased by $9.4 \pm 7.0$ Tg CO yr$^{-2}$ ($p = 0.01$), driven by decreasing anthropogenic ($-5.6 \pm 2.2$ Tg CO yr$^{-2}$, $p < 0.01$) and biomass burning ($-3.8 \pm 4.9$ Tg CO yr$^{-2}$, $p = 0.11$) sources. The trend in biomass burning emissions has a large $p$-value, which means statistical non-significance due to a large year-to-year variation (Coefficient of Variation, CV $= 11.6\%$), especially the peak emissions from Southeast Asia (SEAS) at the end of 2015 caused by the 2015–2016 El Niño event (Yin et al., 2016; Liu et al., 2017). The oceanic and biogenic sources account for only 15% of surface CO emissions with a small inter-annual variability (CV$=2.2\%$ and 8.6%, respectively), therefore they have little effect on the trend of the CO total source. In contrast to declining direct emissions, the CO chemical production is estimated to have remained flat (CV$=1.2\%$), resulting from the compensation between increasing yields from CH$_4$ oxidation ($3.0 \pm 0.4$ Tg CO yr$^{-2}$, $p < 0.01$) and decreasing yields from NMVOCs oxidation ($-3.6 \pm 1.5$ Tg CO yr$^{-2}$, $p < 0.01$).

Three variables determine the global CO sink as discussed in Eq. (1): $k_{CO+OH}(T)$, $[CO]$, and $[OH]$. Tropospheric CO columns measured by MOPITT have declined at a relative rate of $-0.32\pm0.05\%$ $yr^{-1}$ ($p<0.01$) during 2000–2017, highly consistent with the relative trend in the estimated CO sink ($-0.35\pm0.23\%$ $yr^{-1}$, $p<0.01$). This suggests that decreasing CO concentrations are the primary driver of the declining CO sink, and dominate over the influence from the possible changes in OH and reaction rate.

### 3.2.2 Influence of OMI HCHO and GOSAT XCH₄ constraints

Inversions #2 and #3 assimilated OMI HCHO and GOSAT $XCH_4$ in the inverse system to directly constrain the reactants of CO chemical production. These two inversions make a small difference (<10% for a single year and <2% for multiannual mean) in the global CO budget estimates compared to Inversion #1 (Table S5), and all three inversions estimate a slightly smaller CO source than the CO sink in most of the years between 2000 and 2017. However, the three inversions reveal different declining trends (Table 4; Fig. 2). Inversion #2 estimates that the global CO source and sink decreased at the rates of $-7.4\pm13.0$ Tg CO $yr^{-2}$ ($p=0.24$) and $-9.7\pm11.7$ Tg CO $yr^{-2}$ ($p=0.09$), respectively, from 2005 to 2017, slightly slower than Inversion #1-estimated trends of $-10.3\pm12.7$ Tg CO $yr^{-2}$ ($p=0.10$) and $-11.3\pm11.0$ Tg CO $yr^{-2}$ ($p=0.05$) in the same period. The large $p$ values for trends during 2005–2017 indicate a large inter-annual variability, mainly caused by the significant biomass burning emissions from peat fires in Indonesia during the 2015–2016 El Niño event (Sect. 3.3). The slower decline in the CO total source estimated by Inversions #2 and #3 is primarily due to the growing CO production (Fig. 2a), in contrast to the flat CO chemical production estimated by Inversion #1. For example, Inversion #2 estimates that the CO production increased by $10.8\pm5.3$ Tg CO $yr^{-2}$ ($p<0.01$) during 2005–2017, and Inversion #3 estimates a growing CO production by $12.8\pm11.3$ Tg CO $yr^{-2}$ ($p=0.03$) during 2010–2017.

Direct anthropogenic CO emissions are estimated to decline faster in Inversions #2 and #3 (Table 4 – those trends being compared during the same overlapping periods for inversions in this table). Inversion #1 estimates that anthropogenic CO emissions declined by $-6.4\pm3.5$ Tg CO $yr^{-2}$ ($p<0.01$) during 2005–2017, while Inversion #2 shows a steeper decline of $-11.1\pm4.0$ Tg CO $yr^{-2}$ ($p<0.01$). During 2010–2017, Inversions #2 and #3 estimate declining rates of $-14.6\pm5.4$ Tg CO $yr^{-2}$ ($p<0.01$) and $-12.7\pm3.8$ Tg CO $yr^{-2}$ ($p<0.01$), respectively, both faster than the Inversion #1 estimated trend of $-7.5\pm6.5$ Tg CO $yr^{-2}$ ($p=0.03$). For the top 5 emitters of anthropogenic CO (Fig. 2b), we see faster declines in the USA, CHN, and EU and slower growth over SAS and EQAF than those present in Inversion #1. However, biomass burning emissions and trends tend to remain almost unchanged in Inversions #2 and #3 (Table 4; Fig.2c).

### 3.2.3 Best estimate between different inversions

Inversions #1, #2, and #3 all clearly suggest that decreased surface emissions from anthropogenic and biomass burning sources are major drivers of the declining global CO source during 2000–2017, however, they estimate different trends in anthropogenic CO emissions and CO chemical production. This suggests that the additional observation constraints related to the CO reaction sequence alter the inversion-estimated trends of the global CO budget.

Inversion #1 is capable of separating the trend of the CO total source from the trend of the CO total sink, broadly consistent with the results derived from Inversion #2 and Inversion #3, but the contribution of reduced anthropogenic sources to the declining CO emissions seems to be underestimated in Inversion #1 because the increasing CO chemical production is not directly constrained in that inverse configuration. The increased CO chemical production is reflected by the growing HCHO

in the atmosphere, which is an intermediate reaction product in the oxidation of hydrocarbons. Tropospheric HCHO columns as observed by OMI have been reported to keep growing over China, India, and part of the USA over the last decade (De Smedt et al., 2010; Zhu et al., 2017; Shen et al., 2019), probably related to strong increases in anthropogenic NMVOCs emissions. The bias of Inversion #1 that does not constrain the HCHO and CO production is region-dependent, but most evident in anthropogenic source regions (e.g., China and the US) where rapidly increasing man-made NMVOCs emissions

dominate over relatively stable biogenic NMVOCs. With additional OMI HCHO and GOSAT XCH$_4$ constraints, the CO chemical production in these regions is estimated to increase instead of being flat, which further leads to a faster decrease in the estimated anthropogenic CO emissions to maintain the overall declining CO burden in the atmosphere. In biomass burning regions where biogenic NMVOCs emissions are relatively stable, the estimates of biomass burning CO emissions are quite consistent across the three different inversions.

The most realistic inversion estimate of the global CO budget should be the one with sufficient constraints not only on the atmospheric CO abundance but also on the CO chemical production. The CO chemical production is an important term of the CO budget trends, as it experienced a steady increase due to growing HCHO and CH$_4$ concentrations in the atmosphere. Constraining the CO chemical production can correct the inversion system that may inaccurately attribute some of the decreases in the CO source to the CO chemical production. Therefore, it is reasonable to think that Inversion #3 has a more

realistic representation of the source splitting between anthropogenic emissions and chemical production in the global CO budget than Inversion #2 does, and Inversion #2 is better than Inversion #1. It is appropriate to use Inversion #3 and Inversion #2 for the trend analysis but these inversions are limited to short periods. If Inversion #1 has to be used due to its long-term temporal coverage, caution needs to be taken that the decreasing trends of anthropogenic CO emissions are probably underestimated and the increasing trends of CO chemical production are not well separated over anthropogenic source regions.

For the trend analysis in this paper, we present all the estimates from Inversions #1, #2, and #3 for completeness. The global CO budget data derived from all the three inversions can be found at the data repository of https://doi.org/10.6084/m9.figshare.c.4454453.v1 (Zheng et al., 2019).

### 3.2.4 Comparison with the prior CO budget

The comparison to the prior modelled CO budget (Table S6) shows that the inversion system adjusts both the magnitudes and

trends of the CO source and CO sink. Inversions #1, #2, and #3 exhibit similar estimates in the global CO source, which are 15% (~ $3.4 \times 10^2$ Tg CO yr$^{-1}$) larger than the prior CO flux on average, including $1.1 \times 10^2$ Tg CO yr$^{-1}$ from anthropogenic sources, $1.1 \times 10^2$ Tg CO yr$^{-1}$ from biomass burning sources, and $0.9 \times 10^2$ Tg CO yr$^{-1}$ from biogenic sources. The increment of 15% is within the uncertainty range of bottom-up CO inventories, which are typically subject to a one-sigma uncertainty

between 26% and 123% for top emitting anthropogenic regions and countries (Crippa et al., 2018). The inversion-based CO sink is 14% higher than the prior estimates, equivalent to another $3.2\times10^2$ Tg CO $yr^{-1}$ loss. For trends, the prior results exhibit an increasing CO source ($3.6\pm3.8$ Tg CO $yr^{-2}$, $p$=0.07) that could result in a growing atmospheric CO burden from 2000 to 2017, disagreeing with the observed declining CO since 2000. The inversion results reverse the upward trend in prior

anthropogenic emissions to a rapid downward trend and estimate larger decreases in biomass burning emissions.

### 3.3 Regional atmospheric CO budget

### 3.3.1 Regional distribution

The global CO source, the sum of surface emissions and chemical production, follows a bimodal distribution by latitude (Fig. 3a, 4a, S3a, S4a). The highest peak is in tropical regions (30° S–30° N) where 70% of the global CO source is located, including

83% of biomass burning emissions, 75% of CO chemical production, and 50% of anthropogenic emissions. The regions closest to the equator, such as South America, Equatorial Africa, Southeast Asia, and Northern Australia, are responsible for most of the biomass burning emissions (Fig. 5c) and of the CO chemical production. The anthropogenic emission hotspots are Equatorial Africa, South Asia, and South China (Fig. 5a). The other peak latitude band of CO source is the northern mid-latitudes (30° N–60° N), which account for 23% of the global CO source, dominated by the anthropogenic emissions from the

US, Europe, and China (Fig. 5a). On average, 47% of the global anthropogenic CO emissions are distributed within 30° N–60° N.

The global CO sink presents an asymmetrical distribution around the equator that is 30% larger in the Northern Hemisphere than that in the Southern Hemisphere, due to the higher CO levels in the Northern Hemisphere (Fig. 3a, 4c, S3c, S4c). The tropical regions (30° S–30° N) that have both the largest CO source and OH concentration (Lelieveld et al., 2016) account for

71% of the global CO sink, consistent with the proportion of CO source distributed over this region. The northern mid-latitudes (30° N–60° N) account for only 17% of the global CO sink but 23% of the CO source due to a much lower OH concentration than in the tropical troposphere. The southern mid-latitudes (30° S–60° S) account for 9% of global CO sink, corresponding to 5% of the global CO source located in this region. A strong CO sink is also evident near coastlines over the ocean (e.g. west of the African continent) mainly due to the fact that the CO transported out of lands driven by prevailing winds further react

with OH over the ocean.

For trends, the northern mid-latitudes (30° N–60° N) show the sharpest declines in both CO source and CO sink, consistently presented in Inversion #1 (Fig. 3b), #2 (Fig. 3c), and #3 (Fig. 3d). The decline in the CO source is most evident in the US, Europe, and China (Fig. 4b, S3b, S4b), mainly caused by reduced anthropogenic sources (Fig. 5b). This drives the largest regional decrease in CO total columns between 30° N–60° N as seen by MOPITT (Fig. 1) and also leads to a significantly

reduced CO sink. The tropical region (30° S–30° N) exhibits both increased and decreased CO sources over different continents. The increased sources over South Asia and Equatorial Africa are both driven by the exponential growth in anthropogenic emissions (Fig. 5b), which partially offsets the decreasing biomass burning emissions in Equatorial Africa and

South America (Fig. 5d). The increase of the CO chemical production is seen over the whole tropical region, as shown in Figs. 3c and 3d. However, the estimated CO sink lacks statistically significant trends in the majority of the tropical region, and the continents with fast-growing CO sources (e.g., South Asia, Equatorial Africa) lead to increased CO total columns (Fig. 1), although the global background concentrations are rapidly decreasing.

### 3.3.2 Anthropogenic emissions

The top 5 emitters of anthropogenic CO are CHN, SAS, USA, EQAF, and EU, where large amounts of fossil fuel and biofuel are burned in industrial, transportation, and residential facilities. These five regions are estimated to account for more than 60% of the global anthropogenic CO emissions (Tables S7, S8, S9), and can explain more than 80% of the global downward trend from 2000 to 2017 (Fig. 6a). CHN, USA, and EU are estimated to have reduced their emissions rapidly, which more than offsets the increasing emissions from SAS and EQAF. Emissions from all the other regions contribute much less to total anthropogenic emissions, and most of the emission changes are not statistically significant ($p>0.1$), therefore have little influence on the anthropogenic CO emissions trends.

A comparison with the bottom-up anthropogenic inventory CEDS that we use as prior shows that the Inversion #1 estimates stay close to CEDS over CHN, SAS, and the USA, but have larger values in regions with medium-sized emissions (20–50 Tg CO yr$^{-1}$), such as in EU and EQAF (Fig. 7a; Table S7). The inversion-based larger emissions in those regions help correct the biases of the prior modelled CO concentrations with respect to independent surface observations (Sect. 3.4). For the 2000–2017 trend (Fig. 7b), the discrepancy between Inversion #1 and CEDS mainly occurs for the top two emitters, CHN and SAS, although their long-term averages agree well. Inversion #1 shows that CHN emissions decreased and SAS emissions increased modestly, while both of these two regions are allocated a rapid growth in CEDS. As Inversion #1 tends to underestimate the decrease and to overestimate the increase of anthropogenic emissions (Sect. 3.2), the CEDS inventory probably has large biases in emission trends estimates over CHN and SAS, which is the main reason why it estimates growing anthropogenic emissions globally (Table S6) and cannot match the observed declining CO when used in the input of our LMDz-SACS model. This is consistent with the finding of Strode et al. (2016) who performed global CO modelling with a different model and inventory.

### 3.3.3 Biomass burning emissions

Inversions #1, #2, and #3 consistently estimate declining biomass burning CO emissions (Fig. 6b), primarily driven by five regions (EQAF, SAF, Brazil - BRA, SEAS, and RUS) that account for more than 70% of global biomass burning CO (Tables S10, S11, S12). Based on Inversion #1 results, EQAF presents a declining trend of $-1.6\pm1.1$ Tg CO yr$^{-2}$ ($p<0.01$, Table S10) during 2000–2017, while emissions in the other four regions fluctuate. The large inter-annual variability makes the assessment of a trend more uncertain especially given high emissions during extreme drought years. For example, the 2010 emissions are estimated significantly higher than the 2009 emissions due to the suddenly rising emissions in BRA caused by drought in the Amazon forest (Lewis et al., 2011; Xu et al., 2011). The 2015 emissions are estimated close to the maximum since 2000 due to fire anomalies in SEAS and BRA as a consequence of record-breaking drought during the 2015–2016 El Niño event

(Jiménez-Muñoz et al., 2016; Yin et al., 2016; Liu et al., 2017). Besides, the EU and the Korea and Japan (KAJ) both present slightly decreasing biomass burning emissions ($p<0.05$), while Canada (CAN) shows a moderate increasing trend of $0.8\pm0.6$ Tg CO $\text{yr}^{-2}$ ($p<0.05$), concurrent with the increasing burned area (Canadian National Fire Database, 2018). All of the other regions have highly variable biomass burning emissions during 2000–2017 without linear trends ($p>0.1$, Table S10).

The biomass burning CO emissions derived from all three inversions are on average ~30% higher than the GFED 4.1s estimates that we use as prior, mainly because our inversions give ~20%, ~50%, and ~70% higher emissions than GFED for EQAF, SAF, and BRA, respectively (Fig. 7d). The larger biomass burning emissions derived from inversions are most evident in the peak fire month and in late fire seasons when burned area declines after the peak fire month (Fig. S5). This mismatch in biomass burning emission seasonality between bottom-up and top-down estimates is a long-standing problem, especially in

Africa and South America (van der Werf et al., 2006; Roberts et al., 2009; Whitburn et al., 2015; Thonat et al., 2015). Our previous study (Zheng et al., 2018b) suggested that the mismatch over Africa is probably caused by a flaming-to-smoldering transition that occurs in late dry seasons, which increases CO emission factors of savanna fires due to low combustion efficiency and thus push up CO emissions. GFED uses seasonally constant emission factors that represent the mean of measurement mostly for flaming combustions, therefore tends to underestimate the late fire season emissions. Despite being

probably underestimated, GFED estimates consistent regional emission trends (if any) with Inversion #1 (Fig. 7e) because the underestimation bias is canceled when calculating trends.

## 3.4 Evaluation with ground-based observations

Modelled CO columns from Inversions #1, #2, and, #3 all match MOPITT observations within their assigned errors and also in terms of trends (Inversion #1 is shown in Fig. S6). The posterior simulation corrects the underestimates of prior modelled

CO columns, especially over Europe, Africa, and South America, where the CO source is increased by inversion. For trends, the model with prior fluxes can only simulate slightly declining CO columns over the USA and EU (Fig. S6g), where anthropogenic CO emissions decrease in prior, but present increasing CO columns over all the other regions. The inverse system reverses the upward trend in prior CO sources that is inconsistent with atmospheric CO observations (e.g., in Asia), which consequently reproduces the global decline in CO total columns (Figs. S6c, S6e).

The independent ground-based observations from WDCGG and TCCON confirm an improvement of modelled CO and XCO in Inversions #1, #2, and, #3 with respect to both annual averages and trends (Figs. B1–B3, S7–S9). Compared with the WDCGG data, all three inversions correct the underestimates of the prior surface CO concentrations, which reduces the NMB and RMSE and increase the slope (more close to one) and R-squared of the linear regressions. For trends, most of the WDCGG sites, especially those located in the USA, EU, and CHN, present statistically significant downward trends between 2000 and

2017, while the prior results tend to underestimate the declining trends. These biases are reduced by the inversions, though uncertainties still exist in view of the scattered dots. The WDCGG sites that show large disagreements are mostly located at coastal terrain areas, where our coarse-resolution model simplifies the coastline and thus cannot resolve the associated meteorology well (e.g., land-sea breeze circulation) (Palau et al., 2005; Ahmadov et al., 2007) and possible local emission

sources. Several sites at high northern latitudes also suggest relatively large modelling bias due to the lack of high-quality satellite data as an observational constraint. The modelled XCO with posterior fluxes agrees better than the prior results with the TCCON observations, especially for the rapidly declining trends between 2000 and 2017.

The evaluation with measurement from WDCGG suggests that Inversion #3 gives a fair estimate of surface CO trends during
2010–2017 (NMB = −8%, RMSE = 1.4 % yr$^{-1}$, Fig. B3c), while Inversion #2 (NMB = −34%, RMSE = 2.0 % yr$^{-1}$, Fig. B2c) and Inversion #1 (NMB = −47%, RMSE = 1.8 % yr$^{-1}$, Fig. B1c) still present moderate biases in their study period. During the overlap period of 2010–2017 with Inversion #3, Inversion #2 and Inversion #1 both present slightly larger RMSE of 1.5 % yr$^{-1}$ in the trend estimates. Comparing with TCCON observations, we also see a slight improvement of the modelled XCO trends in Inversion #3 (NMB = 21%, RMSE = 0.3 % yr$^{-1}$, Fig. B3d) and in Inversion #2 (NMB = 10%, RMSE = 0.3 % yr$^{-1}$,
Fig. B2d) than those estimated by Inversion #1 (NMB = 9%, RMSE = 0.4 % yr$^{-1}$, Fig. B1d).

## 4 Discussion

### 4.1 Influence of prior inter-annual variation

Inversion #4 has the same inversion set up as Inversion #1 except that it used the flat prior CO fluxes (seasonal climatology) without inter-annual variation. The prior CO fluxes used in Inversion #1 (Table S6) are composed of increasing anthropogenic
emissions (0.33±0.14% yr$^{-1}$, $p<0.01$) and decreasing but highly time-variable biomass burning emissions (−0.43±1.43% yr$^{-1}$, $p=0.53$), which lead to a slightly increasing global total source (0.16±0.18% yr$^{-1}$, $p=0.07$). Without this prior inter-annual variability, Inversion #4 estimates that the global CO source declined by −0.32±0.23% yr$^{-1}$ ($p<0.01$) and the global CO sink declined by −0.32±0.20% yr$^{-1}$ ($p<0.01$) during 2000–2017 (Table 3). The declining CO source can be decomposed into a relatively flat CO chemical production (0.03±0.09% yr$^{-1}$) and a rapidly declining surface emission (−0.62±0.36% yr$^{-1}$) that is
primarily due to the decreasing anthropogenic (−0.70±0.26% yr$^{-1}$) and biomass burning (−0.91±0.81% yr$^{-1}$) emissions. Overall, Inversion #4 is highly consistent with Inversion #1 in regard to the relative trends in the anthropogenic (Table S7) and biomass burning (Table S10) sources globally (Fig. 6) and regionally (Fig. 7), especially for the top five emitters (Figs. 8, 9). This consistency suggests that the estimated declining trends in the global CO source and CO sink are not affected by the inter-annual variations of prior CO fluxes.

### 4.2 Drivers of declining CO emissions

We compare the inversion-based anthropogenic CO emissions to regional bottom-up inventories that are not used in our inversions (Fig. 8). Here we use the bottom-up inventory data from EPA for the USA (USEPA, 2018, Fig. 8a), from MEIC v1.3 for CHN (Zheng et al., 2018c, Fig. 8b), from TNO-MACC III for EU (Kuenen et al., 2014, Fig. 8c), from REAS v2.1 for SAS (Kurokawa et al., 2013, Fig. 8d), and from EDGAR v4.3.2 for EQAF (Crippa et al., 2018, Fig. 8e), while our prior
inventory is the global inventory CEDS (see Section 2.2). These bottom-up emissions data are derived from detailed regional activity maps and sector-specific emission factors, except for EQAF where we use the global inventory EDGAR v4.3.2 (up to

2012) due to the lack of regional inventories there. Figure 8 shows consistent anthropogenic CO emission trends between bottom-up and top-down estimates in all five regions, indicating that the inversion results well capture the inter-annual variation of anthropogenic CO emissions. The sectoral detail of bottom-up data allows us to identify the driving source sector in each region. The transport sector dominates the rapidly decreasing emissions in the USA and EU, while the industrial and

residential sectors drive down the emissions in CHN after 2005. The road transport, industrial, and residential sources all push up emissions in SAS during 2000–2010, while the growing residential source is mainly responsible for the continuous rising emissions in EQAF.

Distinct emission driver sectors reflect different stages of socio-economic development, fuel use, and emission regulation policies in different regions. Developed economies such as the USA and EU have improved their industrial and residential

combustion facilities that now burn relatively clean energy at high combustion efficiencies. The transport sector accounts for the majority of the current emissions, and the progressive pollution control on vehicles has successfully cut CO emissions in the USA (Jiang et al., 2018) and EU (Crippa et al., 2016). CHN and SAS are both on the process of rapid industrialization and urbanization, which made their anthropogenic CO emissions rise up rapidly since 2000, driven by all the source sectors especially the industrial and residential sources. To save energy and reduce air pollution, China has improved the combustion

efficiency and strengthened the end-of-pipe pollution control since 2005, which has successfully cut industrial and residential CO emissions (Zheng et al., 2018a, 2018c). China has also implemented stringent vehicle emission standards, however, the explosive growth in vehicle sales and oil use partly offsets the impact of vehicle pollution control, making the transport sector contribute less to emission reductions. The emissions from SAS are estimated to have increased up to 2010 and have remained flat since then, while the cause of flattening emissions is not very clear due to lack of bottom-up inventories for recent years.

The underdeveloped economies in the EQAF have few emissions from the industrial and transport sources, and most of the emissions increase is driven by the growing residential source mainly for cooking and heating.

Biomass burning emissions are determined by burned area, fuel combustion rate per unit area, and emission factors per unit mass of fuel burnt (van der Werf et al., 2017). Our inversion-based biomass burning CO emissions broadly follow the trends of GFED 4.1s burned area that is derived from satellite observations during 2000–2017 (Fig. 9), which suggests that burned

area is the primary driver of biomass burning emissions variation. The global burned area is observed to have declined since 2000 (Fig. 9a) with the largest decline in the grassland and savanna ecosystems over EQAF (Fig. 9b). This declining trend is primarily driven by agricultural expansion and intensification with more fire management and suppression due to population increase and socioeconomic development (Andela and van der Werf, 2014; Andela et al., 2017). Although extreme drought years could shortly expand the burned area, the overall downward trends in global burned area are robust due to the strong

inverse relationship between fire activities and economic development (Andela et al., 2017). We have also noticed that the variations in burned area and biomass burning emissions did not have perfect matches in some years (e.g., the year of 2015). The mismatch can be explained by the inter-annual variation of fuel combustion rate and emission factors that mostly depend on burning conditions, land cover, and fuel type. For example, the 2015–2016 El Niño event caused severe fires on the drained peatlands in Indonesia that are not burned in normal years. This caused only a moderate increase in burned area but released a

disproportionately large amount of CO because the peat fire emission factor is 210 g CO kg$^{-1}$ dry matter (van der Werf et al., 2017), 3.3 times higher than that of savanna fires (63 g CO kg$^{-1}$ dry matter) that regularly burn every year and contribute roughly half of the global biomass burning CO emissions.

## 4.3 Interactions between OH and CO

The global CO budget is affected by the interactions between OH and CO. A lower OH level translates into a proportionally smaller CO sink and thus estimates a smaller CO total source to achieve the source-sink balance (Müller et al., 2018). As such, the inter-annual variability of OH (if any) perturbs the long-term trends of the global CO budget. There are similar discussions for CH$_4$ that a combination of declining OH and slightly growing CH$_4$ source may explain the renewed growth of CH$_4$ concentrations since 2007 (Rigby et al., 2017; Turner et al., 2017), otherwise an abrupt increase in CH$_4$ emissions since 2007

has to be assumed to match the CH$_4$ observations (Turner et al., 2017). On the other hand, the declining CO burden in the atmosphere can leave more OH available to oxidize CH$_4$ through the coupling of CO-OH-CH$_4$, which means that the declining CO can stimulate the growth of CO chemical production (Gaubert et al., 2017). These interactions between OH and CO help understand the uncertainties induced by OH in our inversion results.

The tropospheric OH mean derived from Inversion #1 is $9.9 \times 10^5$ molec. cm$^{-3}$ and Inversions #2 and #3 both give a mean OH

of $10.0 \times 10^5$ molec. cm$^{-3}$. These values are close to the TransCom prior with less than 1% difference (the prior uncertainty is 5%). This is a medium level compared to the modelled OH of $6.5 \times 10^5$ to $13.4 \times 10^5$ molec. cm$^{-3}$ in the Atmospheric Chemistry and Climate Model Intercomparison Project (ACCMIP) simulations (Voulgarakis et al., 2013). The three inversions all estimate a small inter-annual variation in OH, less than 2% (Fig. S10). Inversion #1 that constrains OH through CO and MCF gives a small positive trend in OH during 2000–2008 ($0.21 \pm 0.14\%$ yr$^{-1}$, P<0.01) and a small negative trend during 2008–2017

($-0.07 \pm 0.06\%$ yr$^{-1}$, P=0.02). Inversions #2 and #3 both estimate slightly increasing OH with the trends of $0.26 \pm 0.25\%$ yr$^{-1}$ (P=0.05) and $0.19 \pm 0.21\%$ yr$^{-1}$ (P=0.06), respectively, during their overlapping period 2010–2017. The estimated growing OH trends are larger than the downward trends estimated by Inversion #1. As Inversions #2 and #3 additionally assimilate HCHO and CH$_4$ that also react with OH, these results suggest that HCHO and CH$_4$ tend to have a stronger constraint on the OH level than CO and MCF assimilated in Inversion #1. It should be noted that the mixing ratios of MCF in the atmosphere are

approaching zero and therefore hardly constrain OH any more (Liang et al., 2017).

The debate on OH variation is still ongoing (Turner et al., 2019; Nisbet et al., 2019). OH was thought to be well buffered in the atmosphere, which means that OH is not sensitive to the variations of anthropogenic and natural emissions. This is reflected in the results of global chemistry and climate models (Naik et al., 2013; Voulgarakis et al., 2013). A recent 3-D inverse modelling by McNorton et al. (2018) also gave a small inter-annual variation of $1.8 \pm 0.4\%$ in the tropospheric OH from 2007

to 2015, broadly consistent with our inversion results (Fig. S10). In contrast, some two-box model inversions give larger inter-annual variations of about 5–8% in the global OH mean (Rigby et al., 2017; Turner et al., 2017; Naus et al., 2019, Fig. S10). Rigby et al. (2017) and Turner et al. (2017) primarily used MCF to infer OH, and they both estimated declining OH from 2004/2005 to 2014/2013. This declining trend is not revealed by 3-D modelling studies, including our inversions here. Turner

et al. (2017) also suggested slightly increasing OH since 2013, which agrees with our estimates of Inversions #2 and #3. However, it should be noted that the uncertainty ranges assessed by those two-box model studies are larger than their estimated OH variations. Our 3-D inversions further suggest that the OH trend is sensitive to the observation constraints used. All these features suggest that the OH inter-annual variation is still underdetermined (Rigby et al., 2017; Turner et al., 2017).

The ability to simulate the nonlinear chemistry of OH is still weak in global models, which is another challenge to understand the OH variation. The LMDz-SACS model adopts a linearized chemical scheme to simulate the hydrocarbon reactions including $CH_4+OH$, $HCHO+OH$, $CO+OH$, and $MCF+OH$. The nonlinear dynamics of the OH chemistry, such as the secondary OH production (Lelieveld et al., 2016) and the interaction of OH with the $NO_x$ chemistry (Miyazaki et al., 2017), is not represented. The negligible computational cost of this configuration for OH motivates it, but we also expect the optimization

of OH through the joint assimilation of $CH_4$, HCHO, CO, and MCF observations to counterbalance the simplicity of the scheme. Alternatively, it would be interesting to sophisticate the scheme by introducing key tracers in the OH chemistry (e.g., tropospheric ozone, NO, and NMVOCs) in the scheme together with prescribed (though uncertain) reaction rates, but we currently lack enough observations to constrain this additional complexity.

The OH trends derived from our inversions may be ambiguous, but we can still speculate that our main conclusion is robust

to the possibly larger variation of OH. If a strong downward OH trend existed as two-box model studies suggested, we could see faster decreases in both global atmospheric CO sink and global atmospheric CO source than our current estimates. Although decreasing rates may be varied, the overall trends and drivers of the estimated global CO budget are not changed.

**4.4 Comparison with previous top-down estimates**

We compile the top-down estimated global CO budget from 12 papers (Table S4) to compare with our inversion results.

Compared to the nine studies using different inversion systems (Fig. 10), our inversion system is the only one with the capability of multi-species constraints and also the only one using MOPITT v7 data to constrain CO, while the previous studies had to use older versions of MOPITT data.

All these top-down studies converge on the estimates of global CO source (Fig. 10a), but disagree on the split between surface emissions and chemical production, and also on the proportions of $CH_4$-based and NMVOCs-based CO productions. For

example, Jiang et al. (2017) estimated consistent declining trends in anthropogenic (Fig. 10d) and biomass burning (Fig. 10e) CO emissions as our estimates, but their annual average emissions are 20–37% lower than our results. The CO production from NMVOCs shows a large spread (Fig. 10h) among different inversion studies. Gaubert et al. (2017) estimated a CO chemical production quite close to our estimates (Fig. 10f), while they gave lower $CH_4$ oxidation and higher NMVOCs oxidation. The biogenic and oceanic CO emissions derived from different inversions are 100–200 Tg CO yr$^{-1}$ and 20 Tg CO

yr$^{-1}$ (Table S4), respectively, which are consistent with our estimates (Table 3). Few studies estimated the global CO sink (Fig. 10b), except Gaubert et al. (2017) who gave a declining CO sink from 2002 to 2013 that agrees with our estimates, but Gaubert et al. (2017) values are 15% lower on average.

The other three inversions in the comparison are all derived from previous versions of our inversion system with multi-species constraints (Fortems-Cheiney et al., 2011, 2012; Yin et al., 2015). Compared to these previous estimates (Fig. S11), we have increased the spatial resolution of our transport model, used improved prior data, and assimilated the new MOPITT v7 observations. Therefore, the updated inversion results in this paper are expected to have better quality, leading to updated

trends estimates of the global CO source (Fig. S11a), CO sink (Fig. S11b), surface CO emissions (Fig. S11c), and the CO production from NMVOCs (Fig. S11h).

## 5 Data availability

The data we use as an input of our inversion models and their references are as follows. The MOPITT v7 TIR-NIR product (Deeter et al., 2017) for 2000–2017 can be downloaded from https://l0dup05.larc.nasa.gov/opendap/MOPITT/MOP02J.007/.

The OMI v3 HCHO column retrievals (González Abad et al., 2015) for 2004–2017 can be downloaded from https://aura.gesdisc.eosdis.nasa.gov/data/Aura_OMI_Level2/OMHCHO.003/. The GOSAT $XCH_4$ retrievals produced by the University of Leicester (Parker et al., 2011) for 2009–2017 can be downloaded from http://www.leos.le.ac.uk/data/GHG/GOSAT/v7.2/CH4_GOS_OCPR_v7.2.tar.gz. The ground-based observations from WDCGG can be downloaded from https://gaw.kishou.go.jp/, and those from TCCON (Wunch et al., 2011) can be downloaded

from https://tccondata.org/. The CEDS emissions data (Hoesly et al., 2018) can be downloaded from http://www.globalchange.umd.edu/ceds/ceds-cmip6-data/. The GFED emissions data (van der Werf et al., 2017) can be downloaded from https://www.globalfiredata.org/.

The global CO budget during 2000–2017 derived from Inversions #1, #2, and #3 in this study can be downloaded from https://doi.org/10.6084/m9.figshare.c.4454453.v1 (Zheng et al., 2019). These are monthly gridded data product at the spatial

resolution of 3.75° longitude × 1.9° latitude, including surface CO emissions from different source sectors (i.e., anthropogenic, biomass burning, biogenic, and oceanic), CO chemical production, and CO chemical sink.

## 6 Conclusions

We have estimated the global atmospheric CO budget during 2000–2017 through a multi-species atmospheric inversion system, and have investigated its magnitude, variation, and drivers to understand the observed steady decline in the atmospheric

CO burdens. The inversion-based CO budget significantly improves the modelled CO concentrations and trends compared with independent ground-based observations from the WDCGG and TCCON archives. The inversion results attribute the drivers of the declining MOPITT CO columns during 2000–2017 to a decrease in anthropogenic and biomass burning CO emissions that more than offsets the growing CO chemical production in the atmosphere. The decline in anthropogenic CO emissions mainly occurs in the US, Europe, and China, highly consistent with state-of-the-art regional bottom-up inventories,

which show that the transport sector drives emissions down in the US and Europe, and the improved combustion efficiency

and pollution control in the industrial and residential sources are major drivers in China. The declining biomass burning emissions are consistent with the overall downward trends of satellite-based global burned areas, which is the consequence of agricultural and economic expansion as reported in other published studies. Nonetheless, biomass burning still has significant inter-annual variability and releases a large amount of CO in extreme drought years. We have investigated the robustness and uncertainties of the inversion results through three inversion analyses and one sensitivity inversion tests, from which we demonstrated that the overall declining trends in global and regional CO sources and the underlying drivers are robust to different observation constraints, to prior inter-annual variation, and to possible OH trends. Additionally, our inversion results include emission estimates for methane and formaldehyde that will be the topic of a future dedicated evaluation.

## Author contributions

BZ, FC, and PC designed the study. BZ performed the inversion analysis of the global CO budget and created the data product. The manuscript was written by BZ and revised and discussed by all the coauthors.

## Competing interests

The authors declare that they have no conflict of interest.

## Acknowledgements

We acknowledge the NCAR MOPITT group for the production of the CO retrievals, the Goddard Earth Sciences Data and Information Services Center for the production of the SAO OMI HCHO retrievals. We thank the WDCGG and TCCON archives to publish the ground-based CO observations, and we are grateful to all the people involved in maintaining the network and archiving the observation data. This work benefited from HPC resources from GENCI-TGCC (Grant 2018-A0050102201). We also thank F. Marabelle for computing support at LSCE. R. J. Parker is funded via the UK National Centre for Earth Observation (NCEO grant number: nceo020005). The GOSAT XCH$_4$ retrievals are processed using the ALICE High Performance Computing Facility at the University of Leicester.

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

**Table 1. Configurations of the atmospheric inverse system.**

| Model setup | Configuration | Main reference |
|---|---|---|
| **Inversion general set up** | | |
| Spatial scale | Global | |
| Spatial resolution | 3.75° longitude × 1.9° latitude | / |
| E-folding correlation length | 1000 km over ocean and 500 km over land | Chevallier et al. (2005) |
| Data assimilation window | 14 months for each year (Nov to Dec) | / |
| Time resolution (emission flux) | 8 days | Yin et al. (2015) |
| Minimizer of cost function | M1QN3 | Gilbert and Lemaréchal (1989) |
| **Chemistry-transport model ($H$)** | | |
| Model name | LMDz-SACS | Pison et al. (2009) |
| Meteorological forcing | Nudged to ECMWF ERA-Interim | Dee et al. (2011) |
| Spatial resolution | 3.75° longitude × 1.9° latitude × 39 layers | |
| Convection scheme | Tiedtke's scheme | Tiedtke (1989) |
| **Control vector ($x$)** | Gridded emissions of CO, $CH_4$, and methyl chloroform (MCF); 2D gridded scaling factors to the HCHO production from NMVOCs; 2D gridded scaling factors to the initial concentrations of CO, $CH_4$, MCF, and HCHO in the first time step; scaling factors to OH for 6 big regions globally | Yin et al. (2015) |
| **Prior information ($x^b$)** | | |
| CO emissions | Anthropogenic source: CEDS | Hoesly et al. (2018) |
| | Biomass burning: GFED 4.1s | van der Werf et al. (2017) |
| | Biogenic source: MEGAN | Sindelarova et al. (2014) |
| | Ocean source: POET | Olivier et al. (2003) |
| $CH_4$ emissions | Anthropogenic source: CEDS | Hoesly et al. (2018) |
| | Biomass burning: GFED 4.1s | van der Werf et al. (2017) |
| | Wetland: WetCHARTs | Bloom et al. (2017) |
| | Other $CH_4$ sources | Saunois et al. (2016) |
| MCF emissions | Derived from our previous work | Yin et al. (2015) |
| HCHO from NMVOCs | Pre-calculated by the full chemistry model LMDz-INCA | Folberth et al. (2006) |
| Model initial state | Produced by the LMDz-INCA model | Folberth et al. (2006) |
| OH fields | 3D OH fields from TransCom | Patra et al. (2011) |
| **Observation vector ($y$)** | | |
| CO total column | MOPITT v7 TIR-NIR product (available since Mar 2000) | Deeter et al. (2017) |
| HCHO total column | OMI version 3 (available since Oct 2004) | González Abad et al. (2015) |
| $XCH_4$ | GOSAT retrievals produced by the University of Leicester (available since Apr 2009) | Kuze et al. (2009), Parker et al. (2011) |
| MCF concentration | Surface observations from WDCGG | https://gaw.kishou.go.jp/ |

**Table 2. Atmospheric inversions performed in this work.**

| # | Time period | Prior emissions | Observation constraints |
|---|---|---|---|
| 1 | 2000–2017 | Time-variant data as described in Table 1. | MOPITT v7 CO total column and WDCGG MCF concentrations. |
| 2 | 2005–2017 | Same as Inversion #1 but for 2005–2017. | Observation constraints of Inversion #1 in addition with OMI v3 HCHO total column. |
| 3 | 2010–2017 | Same as Inversion #1 but for 2010–2017. | Observation constraints of Inversion #2 in addition with GOSAT $XCH_4$. |
| 4 | 2000–2017 | Same as Inversion #1 except that prior CO fluxes are the 2000–2017 annual average without inter-annual variation. | Same as Inversion #1. |

**Table 3. Global atmospheric carbon monoxide budget during 2000–2017.** Average CO budget ($10^3$ Tg CO yr$^{-1}$), coefficient of variation (CV, %), and absolute trends from 2000 to 2017 (Tg CO yr$^{-2}$) are derived from Inversion #1 (see Table 2). Relative trends (% yr$^{-1}$) with 95% confidence limits are shown for both Inversions #1 and #4 (no inter-annual variation in the prior CO flux). Significant trends are marked by asterisks (*$p < 0.1$, ** $p < 0.05$, and *** $p < 0.01$).

| | Average ($10^3$ Tg CO yr$^{-1}$) | CV (%) | Absolute trend (Inversion #1) (Tg CO yr$^{-2}$) | Relative trend (Inversion #1) (% yr$^{-1}$) | Relative trend (Inversion #4) (% yr$^{-1}$) |
|---|---|---|---|---|---|
| **Sources** | | | | | |
| Anthropogenic | 0.7 | 5.0 | $-5.6 \pm 2.2$*** | $-0.69 \pm 0.27$*** | $-0.70 \pm 0.26$*** |
| Biomass burning | 0.5 | 11.6 | $-3.8 \pm 4.9$ | $-0.91 \pm 1.15$ | $-0.91 \pm 0.81$** |
| Oceanic | 0.02 | 2.2 | $-0.05 \pm 0.04$** | $-0.22 \pm 0.18$** | $-0.31 \pm 0.28$** |
| Biogenic | 0.2 | 8.6 | $0.1 \pm 1.6$ | $0.04 \pm 0.84$ | $0.20 \pm 0.76$ |
| **Sub-total direct emissions** | 1.4 | 6.1 | $-9.4 \pm 7.0$** | $-0.65 \pm 0.48$** | $-0.62 \pm 0.36$*** |
| Oxidation of CH$_4$ | 0.9 | 1.8 | $3.0 \pm 0.4$*** | $0.35 \pm 0.04$*** | $0.33 \pm 0.04$*** |
| Oxidation of NMVOCs | 0.3 | 7.2 | $-3.6 \pm 1.5$*** | $-0.94 \pm 0.39$*** | $-0.65 \pm 0.33$*** |
| **Sub-total chemical oxidation** | 1.2 | 1.2 | $-0.6 \pm 1.4$ | $-0.05 \pm 0.12$ | $0.03 \pm 0.09$ |
| **Total sources** | 2.6 | 3.3 | $-10.0 \pm 6.9$*** | $-0.37 \pm 0.26$*** | $-0.32 \pm 0.23$*** |
| **Sinks** | | | | | |
| OH reaction | 2.6 | 2.9 | $-9.3 \pm 6.0$*** | $-0.35 \pm 0.23$*** | $-0.32 \pm 0.20$*** |

**Table 4. Absolute trends in global atmospheric carbon monoxide budget.** Absolute trends (Tg CO yr$^{-2}$) with 95% confidence limits are estimated for the time period of 2005–2017 and 2010–2017 using Inversions #1, #2, and #3 (see Table 2). Significant trends are marked by asterisks (*$p < 0.1$, ** $p < 0.05$, and *** $p < 0.01$).

| Unit: Tg CO yr$^{-2}$ | Inversion #1 2005–2017 | Inversion #1 2010–2017 | Inversion #2 2005–2017 | Inversion #2 2010–2017 | Inversion #3 2010–2017 |
|---|---|---|---|---|---|
| **Sources** | | | | | |
| Anthropogenic | −6.4 ± 3.5*** | −7.5 ± 6.5** | −11.1 ± 4.0*** | −14.6 ± 5.4*** | −12.7 ± 3.8*** |
| Biomass burning | −4.4 ± 8.5 | −0.8 ± 20.8 | −4.8 ± 7.8 | −2.3 ± 19.4 | −3.1 ± 19.0 |
| Oceanic | 0.0 ± 0.1 | 0.0 ± 0.2 | 0.0 ± 0.1 | −0.3 ± 0.2** | −0.3 ± 0.3** |
| Biogenic | −0.2 ± 2.9 | 1.4 ± 5.8 | −2.4 ± 1.9** | −2.5 ± 3.3 | −3.0 ± 4.7 |
| **Sub-total direct emissions** | −11.1 ± 12.6* | −6.9 ± 29.5 | −18.2 ± 10.6*** | −19.7 ± 21.0* | −19.0 ± 22.8* |
| Oxidation of CH$_4$ | 3.3 ± 0.6*** | 4.3 ± 1.4*** | 6.1 ± 2.0*** | 9.6 ± 4.0*** | 2.8 ± 6.4 |
| Oxidation of NMVOCs | −2.5 ± 2.3** | −4.1 ± 6.9 | 4.7 ± 4.0** | 11.0 ± 10.0** | 10.0 ± 8.6*** |
| **Sub-total chemical oxidation** | 0.8 ± 2.2 | 0.1 ± 6.9 | 10.8 ± 5.3*** | 20.6 ± 11.8*** | 12.8 ± 11.3** |
| **Total sources** | −10.3 ± 12.7 | −6.8 ± 29.8 | −7.4 ± 13.0 | 0.9 ± 28.6 | −6.2 ± 29.1 |
| **Sinks** | | | | | |
| OH reaction | −11.3 ± 11.0** | −8.6 ± 23.5 | −9.7 ± 11.7* | −0.7 ± 25.0 | −8.9 ± 26.4 |

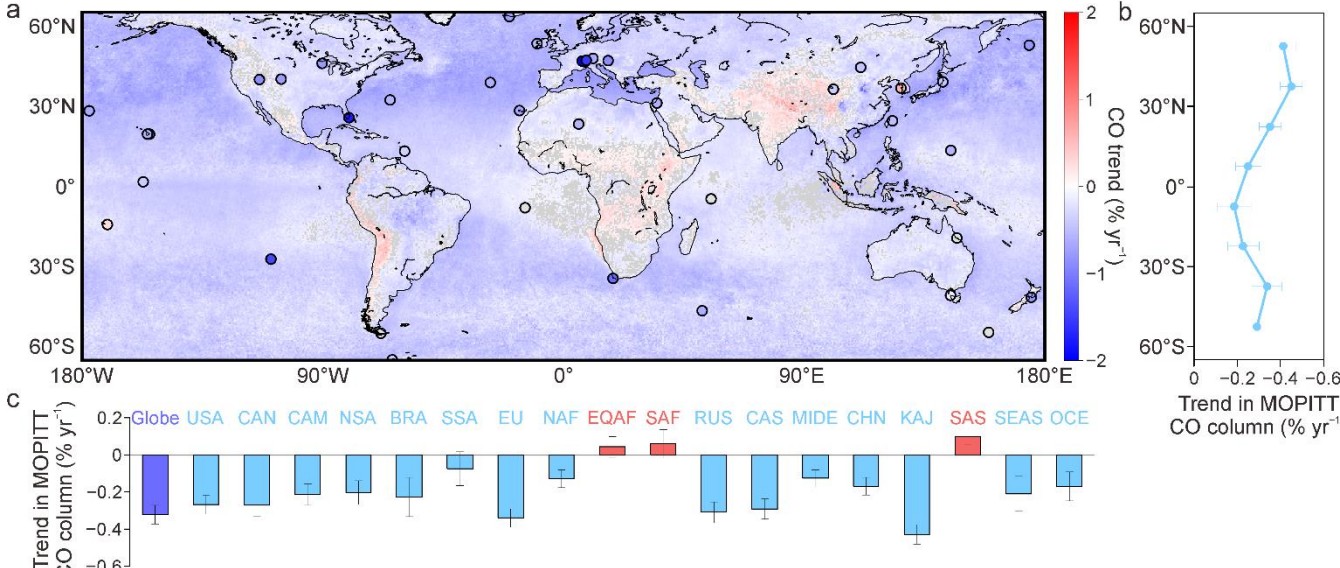

**Figure 1. Trends in the abundance of atmospheric CO from 2000 to 2017.** The map (a) shows the 2000–2017 trends in MOPITT CO total columns at the spatial resolution of $0.5° \times 0.5°$ with the WDCGG sites (dots) that have a continuous measurement during 2000–2017. The colour of the WDCGG dots represent the trends in surface CO concentrations. The curve in (b) shows the trends in MOPITT CO columns by latitude band. The bars in (c) show regional CO column trends (region split refers to Fig. A1). The trends in (a), (b), and (c) are all estimated on the base of monthly time series using a curve fitting method as described in Zheng et al. (2018a). The grey colour in the map (a) indicates the areas or dots without statistically significant trends ($p \geq 0.05$), and the error bars in (b) and (c) represent the 95% confidence intervals. Fig. S1 presents the trends of MOPITT CO columns from 2005 to 2017 (Fig. S1a) and from 2010 to 2017 (Fig. S1b), respectively.

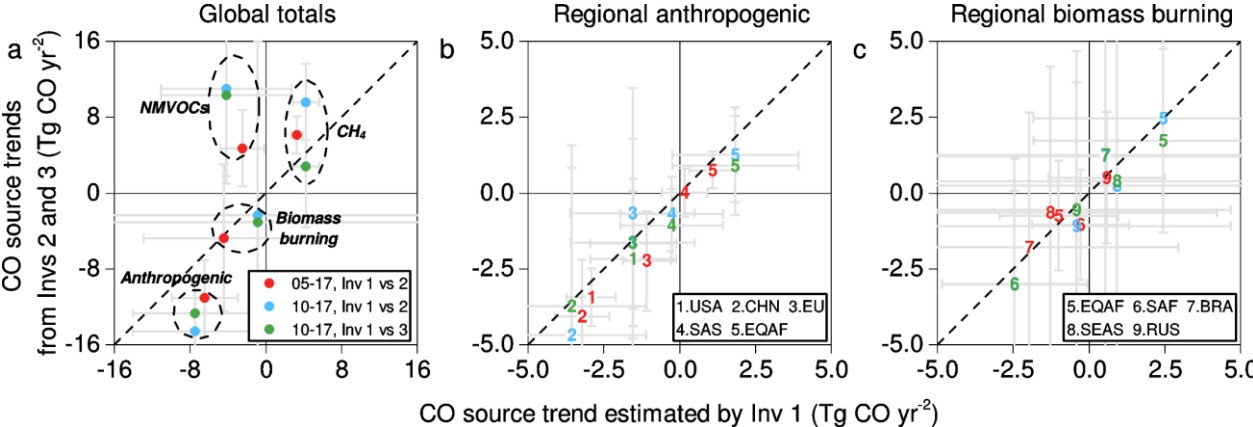

**Figure 2. Comparison of inversion-based CO source trends between Inversions #1, #2, and #3.** The comparison is conducted between Inversions #1 and #2 for 2005–2017 (red dot), between Inversions #1 and #2 for 2010–2017 (blue dot), and between Inversions #1 and #3 for 2010–2017 (green dot). The global totals are presented in (a), and the top 5 emitters (region definition refers to Fig. A1) of regional anthropogenic and biomass burning emissions are presented in (b) and (c), respectively. In all these figures, the CO source trends derived from Inversion #1 are presented along with the x-axis, and the CO source trends from Inversions #2 and #3 are presented along with the y-axis. The error bars for x and y (grey lines) are 95% confidence intervals of the estimated linear trends.

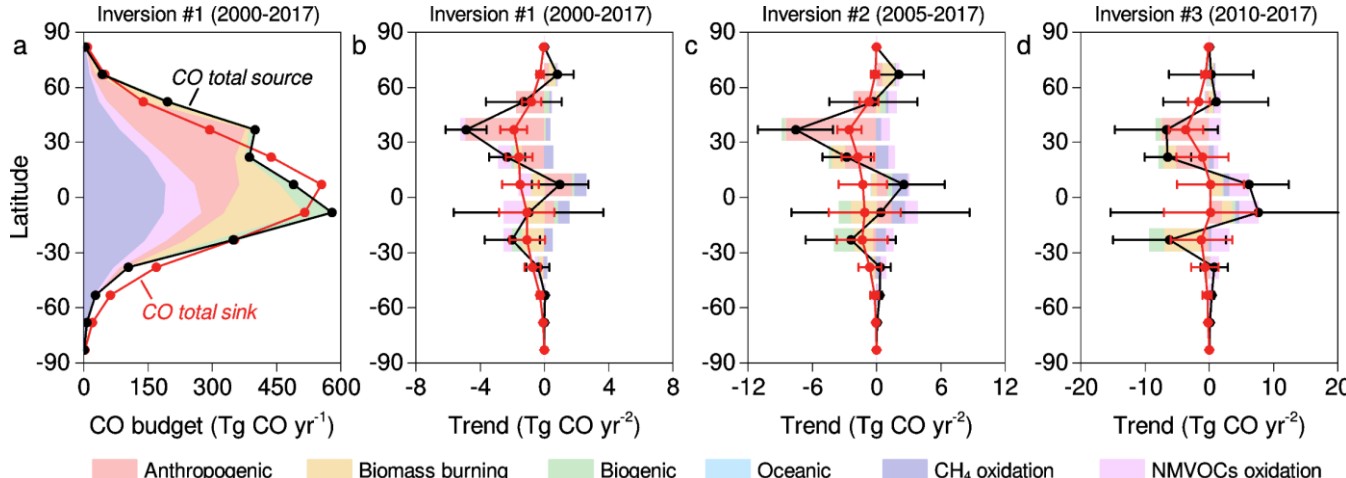

**Figure 3. Global CO budget and trends by latitude band.** The global CO source (black curve and stacked chart) and sink (red curve) derived from Inversion #1 are presented in (a) for every 15-degree latitude band. The budget trends with 95% confidence intervals are estimated for 2000–2017 using Inversion #1 (b), for 2005–2017 using Inversion #2 (c), and for 2010–2017 using Inversion #3 (d). The trends are estimated using the linear least squares fitting method based on annual time series for each 15-degree latitude band.

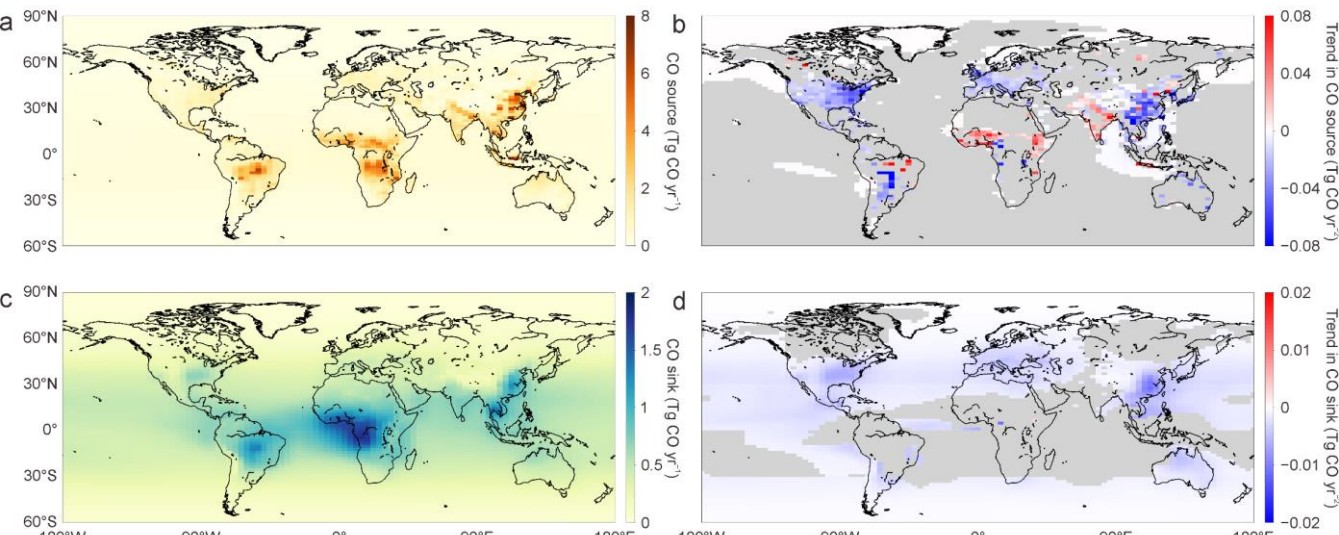

**Figure 4. Spatial distribution of the global CO budget and 2000–2017 trends.** Annual average CO total source and sink during 2000–2017 are shown at the spatial resolution of 3.75° longitude × 1.9° latitude in (a) and (c), respectively, and linear trends of each grid cell are shown in (b) and (d), which are estimated using the linear least squares fitting method based on annual time series. Grey colour in (b) and (d) indicates the areas without statistically significant trends ($p \geq 0.05$). All data shown in this figure are derived from Inversion #1 results. The spatial-temporal distributions derived from Inversion #2 and Inversion #3 are shown in Fig. S3 and Fig. S4, respectively.

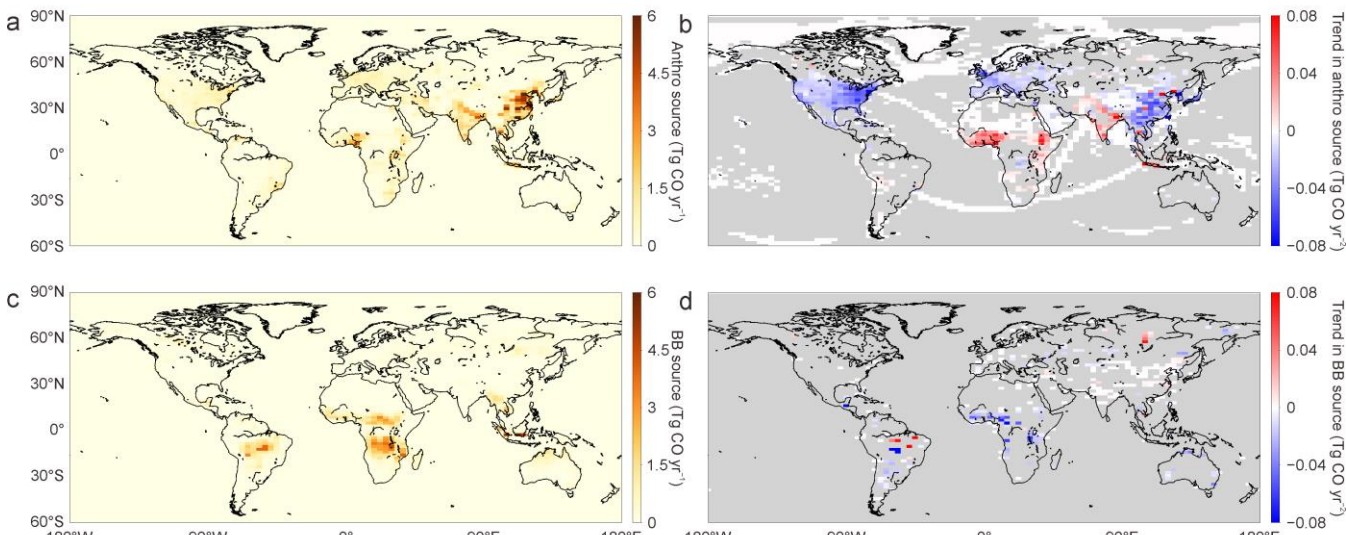

**Figure 5. Spatial distribution of anthropogenic and biomass burning CO emissions and the 2000–2017 trends.** Annual average CO emissions from anthropogenic and biomass burning sources are shown at the spatial resolution of 3.75° longitude × 1.9° latitude in (a) and (c), respectively, and linear trends of each grid cell are shown in (b) and (d), which are estimated using the linear least squares fitting method based on annual time series. Grey colour indicates the areas without anthropogenic and biomass burning emissions or without statistically significant trends ($p \geq 0.05$). All data shown in this figure are derived from Inversion #1 results.

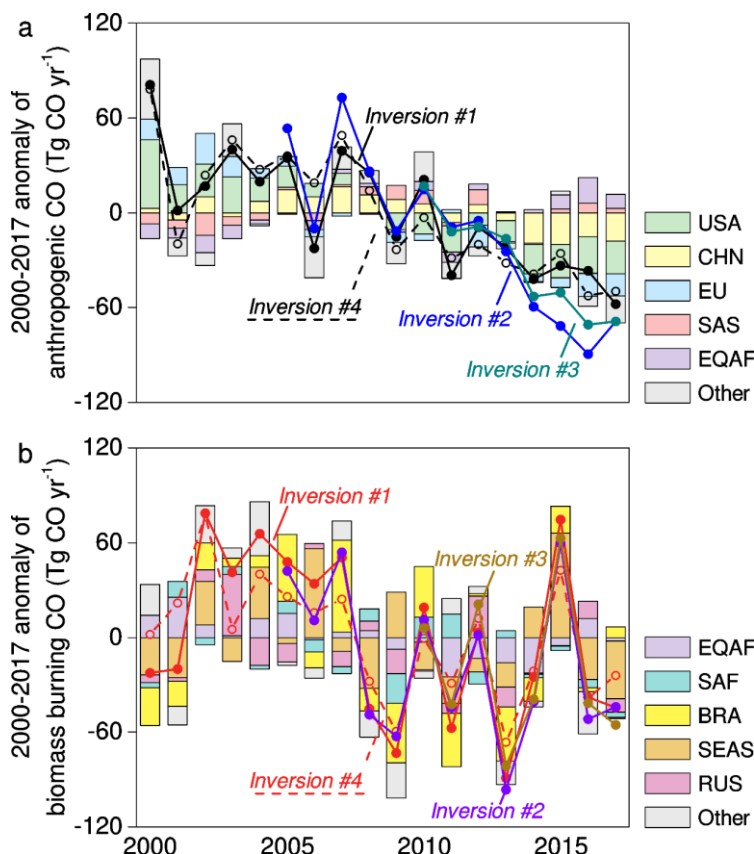

**Figure 6. Global CO emission anomalies from 2000 to 2017.** Anthropogenic (a) and biomass burning (b) emission anomalies are presented with global totals derived from Inversions #1 – #4 (curves), as well as regional emission anomalies (stacked bar) derived from Inversion #1, including the top 5 emitters and the sum of all the other regions. For Inversion #1 and Inversion #4, the emission anomalies are calculated through removing the 2000–2017 average calculated from their own time series data. For Inversion #2 and Inversion #3, the emission anomalies are calculated through removing the 2000–2017 average calculated from Inversion #1.

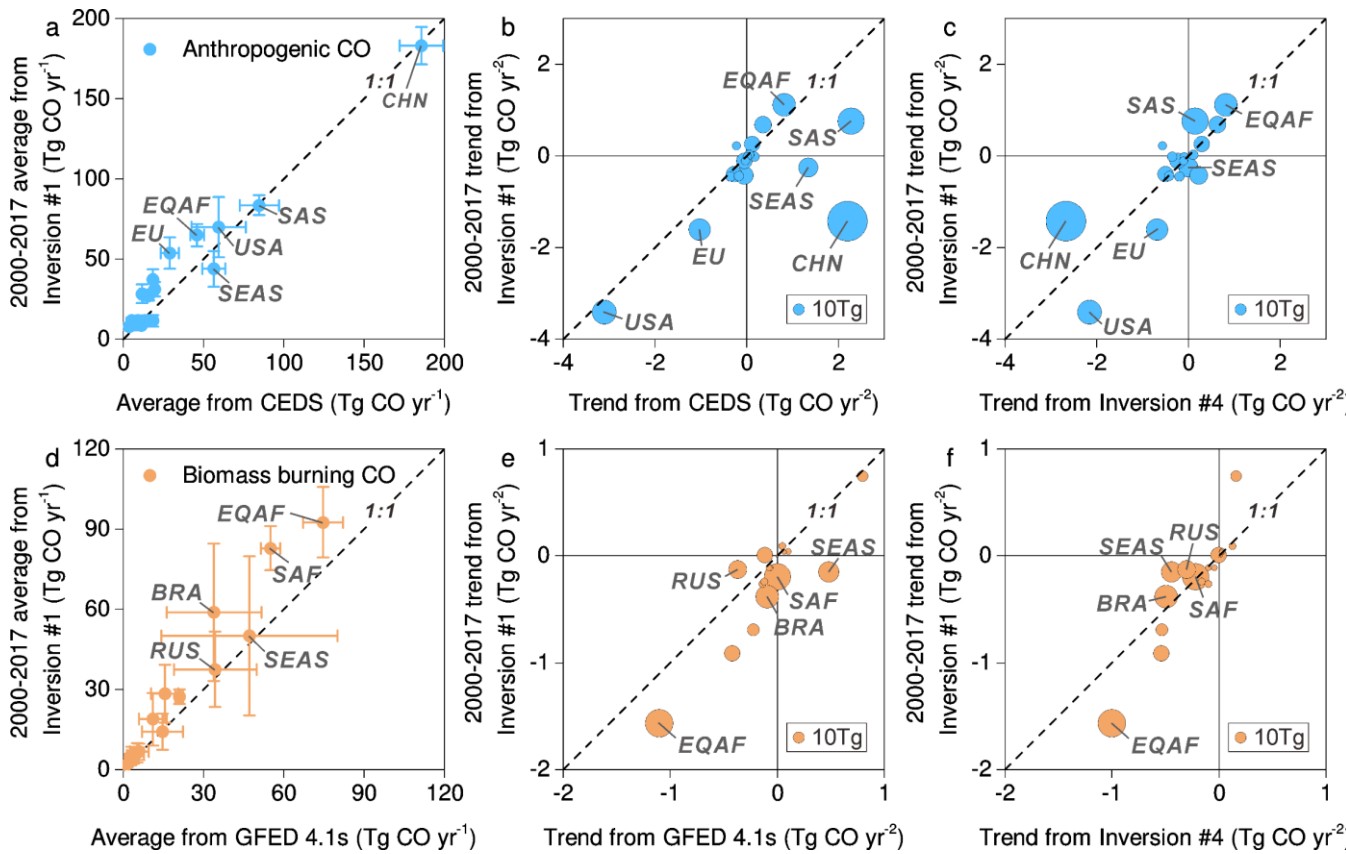

**Figure 7. Comparison between Inversion #1 emission estimates with the prior emissions and Inversion #4 estimates.** Annual average regional anthropogenic emissions (Tg CO yr$^{-1}$, blue dots in a) are compared between Inversion #1 (y-axis) and the CEDS estimate (x-axis) with coefficients of variation as error bars. The linear trends (Tg CO yr$^{-2}$) in regional anthropogenic emissions (blue dots in b and c) are compared between Inversion #1 (y-axis in b and c) and the CEDS inventory (x-axis in b) and Inversion #4 estimate (x-axis in c), respectively, with the area of each dot proportional to annual average emissions derived from Inversion #1. Figures (d), (e), and (f) are similar to (a), (b), and (c), respectively, but for biomass burning CO emissions.

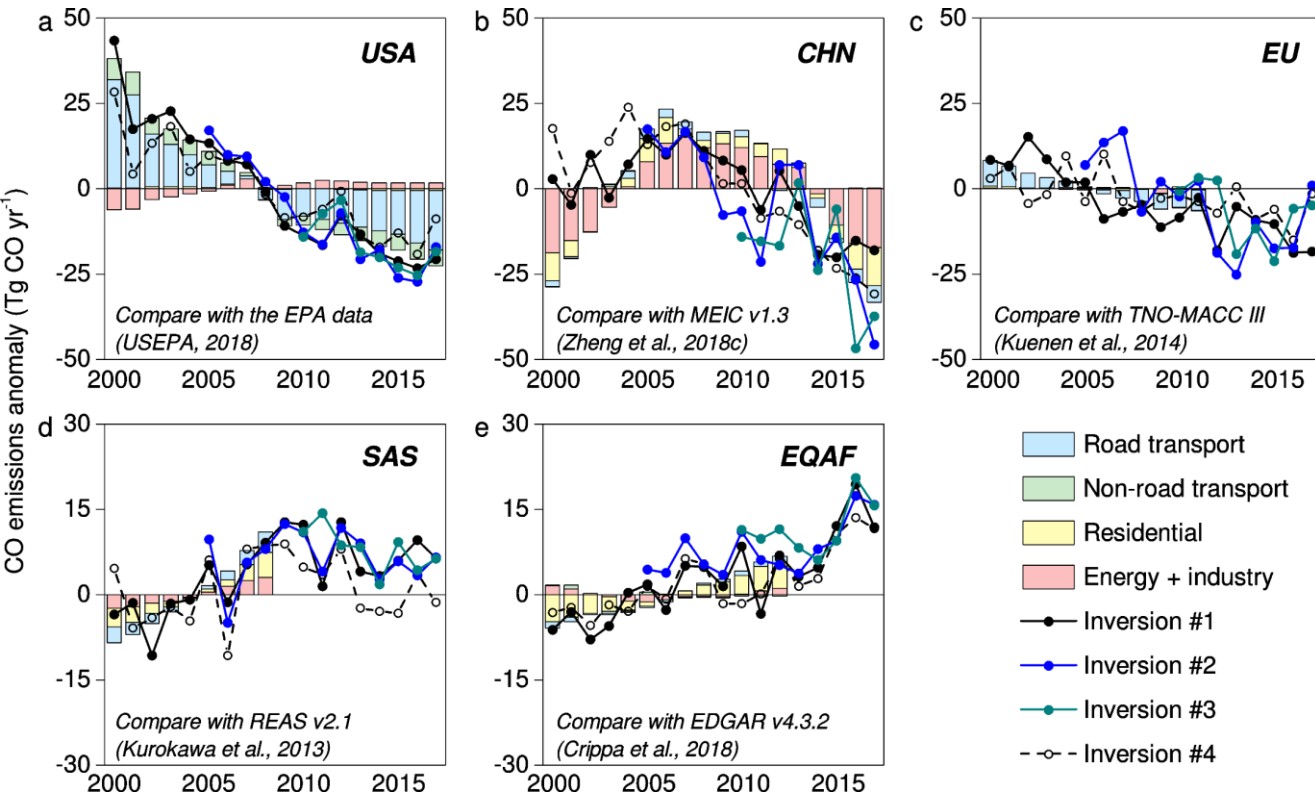

**Figure 8. Comparison of inversion-based anthropogenic CO emission anomalies with bottom-up emission inventories.** The top 5 emitters of anthropogenic CO are presented here including the USA (a), CHN (b), EU (c), SAS (d), and EQAF (e). We compare the regional emission anomalies estimated from Inversions #1 – #4 (curves) to the bottom-up emission inventories that have sectoral details (stacked bar). We normalize emission time series through removing their own annual average, except that Inversion #2 and Inversion #3 subtract the 2000–2017 annual average emissions calculated from Inversion #1.

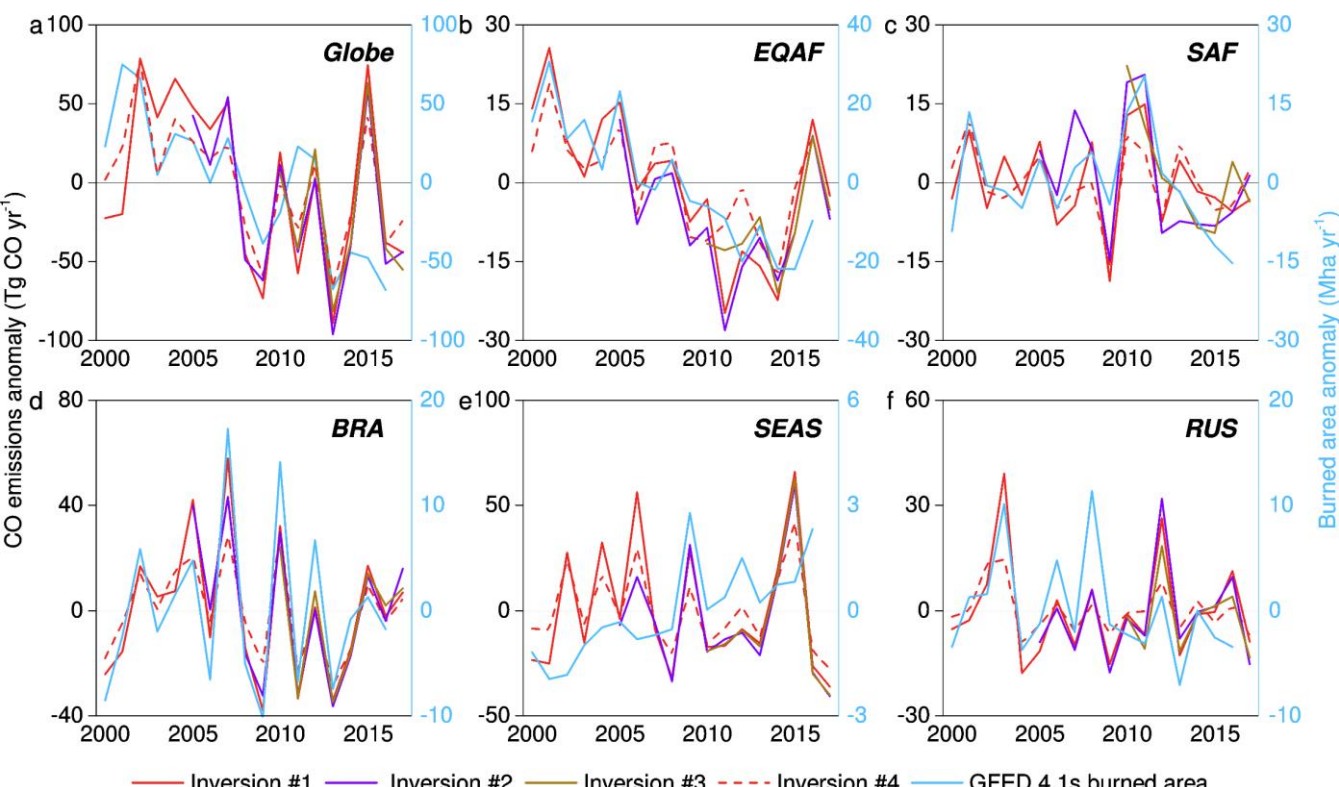

**Figure 9. Comparison of inversion-based biomass burning CO emission anomalies with GFED 4.1s burned area.** Global total (a) and the top 5 emitters of biomass burning CO are presented here including EQAF (b), SAF (c), BRA (d), SEAS (e), and RUS (f). We compare the regional emission anomalies estimated from Inversions #1 – #4 to the GFED 4.1s burned area. We normalize burned area and emissions time series by removing the annual average to calculate the 2000–2017 anomalies, except that Inversion #2 and Inversion #3 subtract the 2000–2017 annual average emissions calculated from Inversion #1.

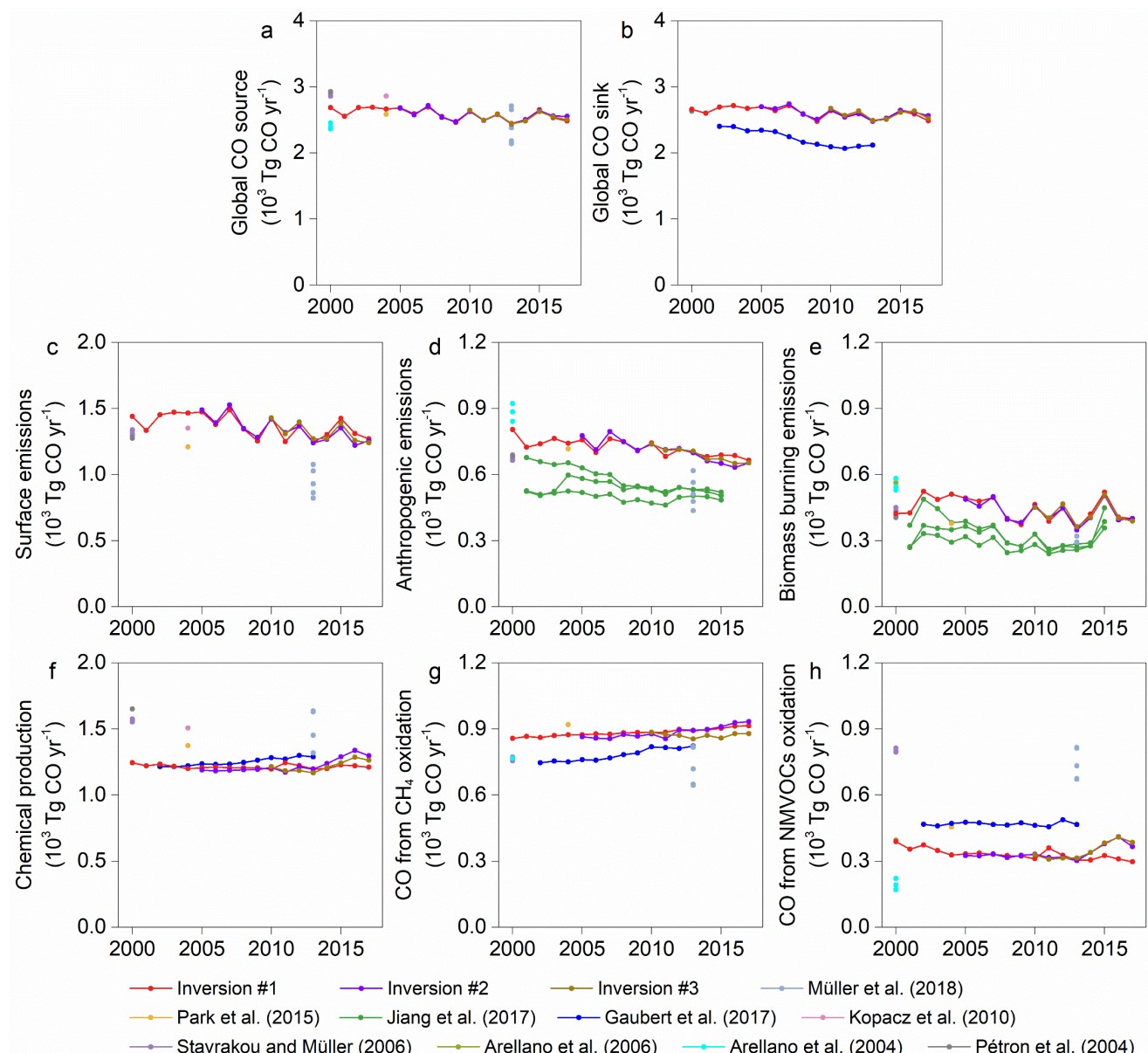

**Figure 10. Comparison of Inversions #1, #2, and #3 with previous top-down estimates of the global CO budget.** The comparison is conducted for the global CO source (a), the global CO sink (b), the surface direct emissions (c), the anthropogenic emissions (d), the biomass burning emissions (e), the CO chemical production (f), the CO production from $CH_4$ oxidation (g), and the CO production from NMVOCs oxidation (h).

## Appendix A: Region splitting in this study

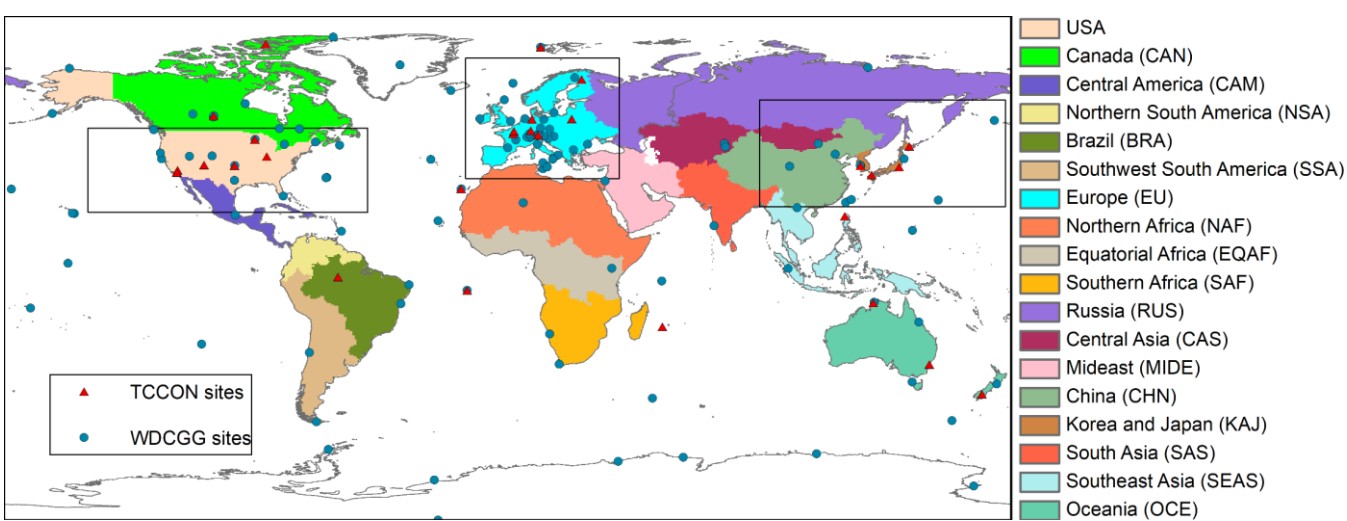

**Figure A1. The 18 regions used in this study.** The WDCGG (blue dot) and TCCON (red triangle) sites are located on the map. The three black boxes select the WDCGG sites used in the model evaluation for the US, Europe, and China, respectively, in Figs. S7, S8, and S9. The TCCON stations include Indianapolis (Iraci et al., 2017a), Manaus (Dubey et al., 2017a), Sodankylä (Kivi et al., 2017), Lauder (Sherlock et al., 2017a, 2017b), Burgos (Morino et al., 2018), Ascension Island (Feist et al., 2017), Réunion Island (De Mazière et al., 2017), Caltech (Wennberg et al., 2017a), Zugspitze (Sussmann and Rettinger, 2018), Ny Ålesund (Notholt et al., 2017a), Orléans (Warneke et al., 2017), Jet Propulsion Laboratory (Wennberg et al., 2017b, Wennberg et al., 2017c), Saga (Kawakami et al., 2017), Izana (Blumenstock et al., 2017), Edwards (Iraci et al., 2017b), Garmisch (Sussmann and Rettinger, 2017), Bremen (Notholt et al., 2017b), Karlsruhe (Hase et al., 2017), Four Corners (Dubey et al., 2017b), Wollongong (Griffith et al., 2017a), East Trout Lake (Wunch et al., 2017), Paris (Té et al., 2017), Anmeyondo (Goo et al., 2017), Park Falls (Wennberg et al., 2017d), Lamont (Wennberg et al., 2017e), Bialystok (Deutscher et al., 2017), Rikubetsu (Morino et al., 2017a), Eureka (Strong et al., 2018), Tsukuba (Morino et al., 2017b), and Darwin (Griffith et al., 2017b).

**Appendix B: Evaluation of posterior simulations of Inversions #1, #2, and #3**

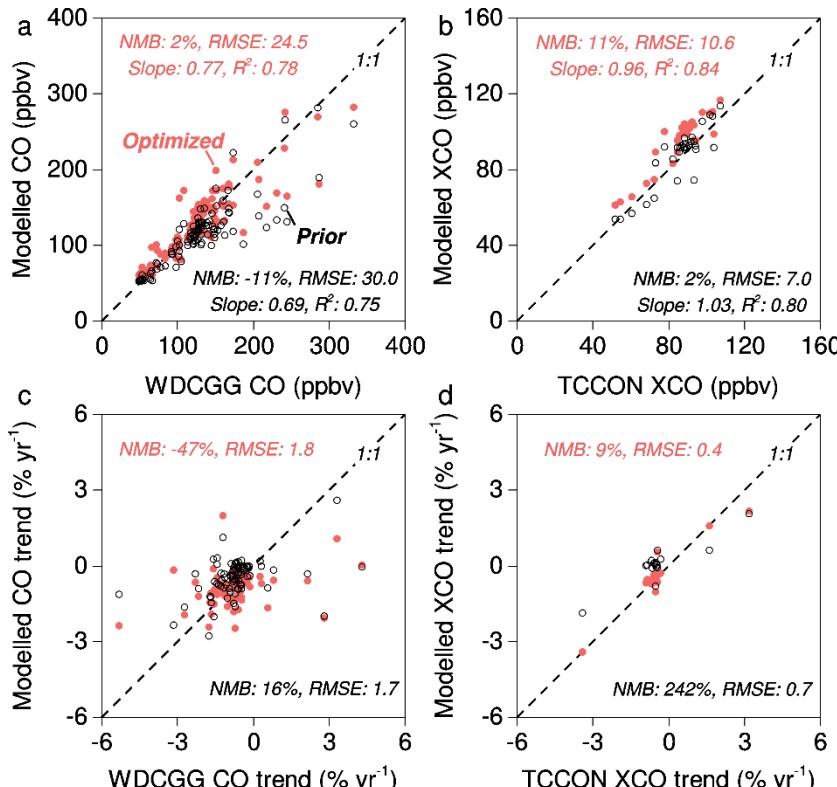

**Figure B1. Evaluation of Inversion #1 with independent ground-based observations.** Annual average surface CO concentrations and XCO modelled by both prior (black dot) and optimized (red dot) emissions are compared with ground-based observations from the WDCGG (a) and TCCON networks (b), respectively. The 2000–2017 trends that are significant in a statistical test ($p < 0.05$) observed by WDCGG (c) and TCCON (d) are used to evaluate the modelled trends. The trends are calculated based on monthly time series using a curve fitting method as described in Zheng et al. (2018a).

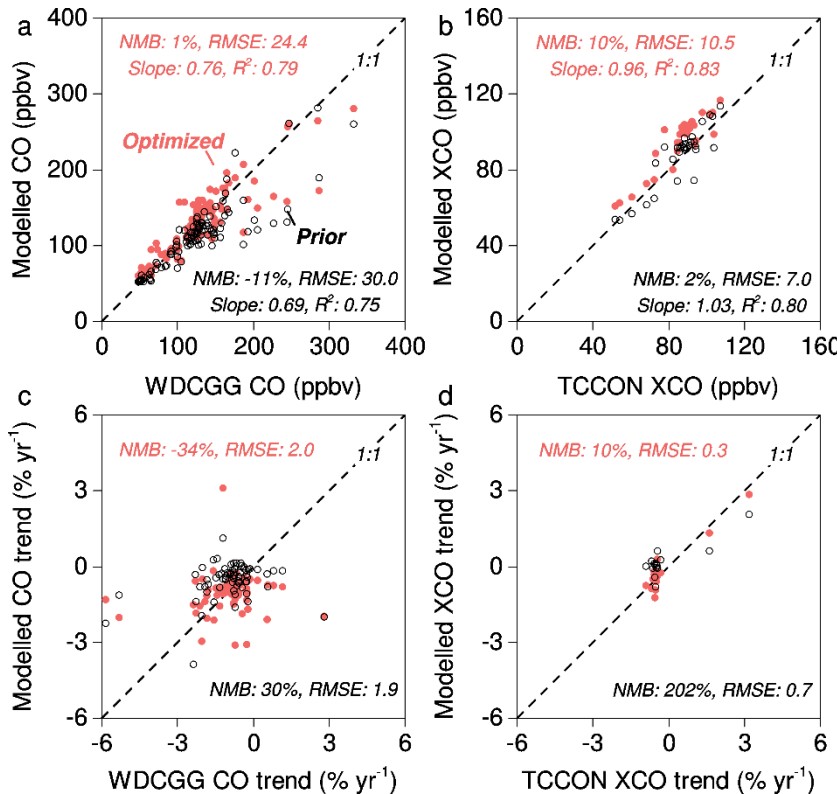

**Figure B2. Evaluation of Inversion #2 with independent ground-based observations.** Annual average surface CO concentrations and XCO modelled by both prior (black dot) and optimized (red dot) emissions are compared with ground-based observations from the WDCGG (a) and TCCON networks (b), respectively. The 2005–2017 trends that are significant in a statistical test ($p < 0.05$) observed by WDCGG (c) and TCCON (d) are used to evaluate the modelled trends. The trends are calculated based on monthly time series using a curve fitting method as described in Zheng et al. (2018a).

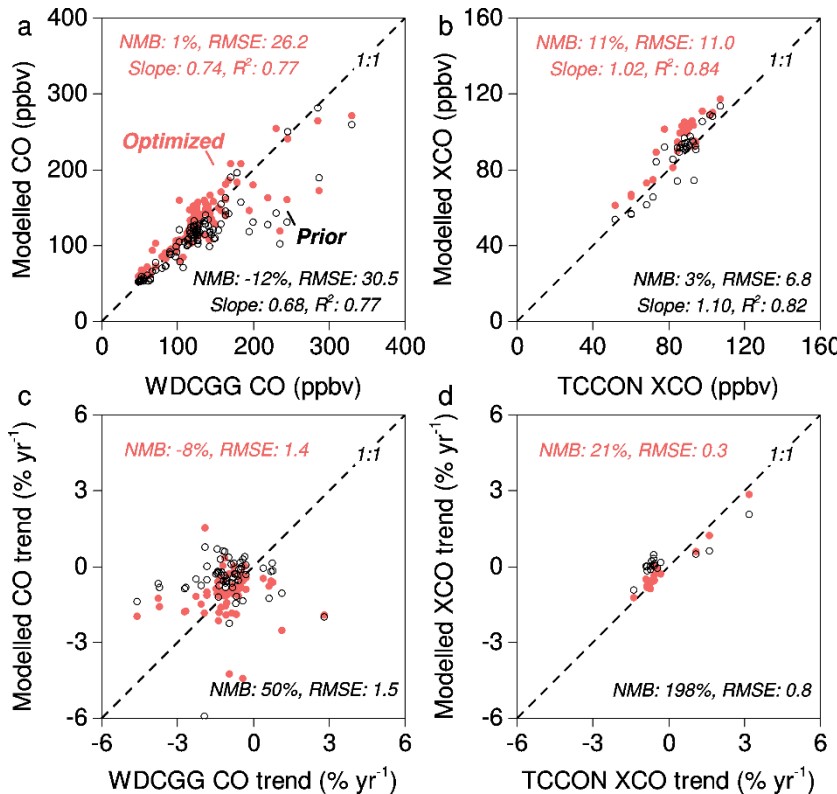

**Figure B3. Evaluation of Inversion #3 with independent ground-based observations.** Annual average surface CO concentrations and XCO modelled by both prior (black dot) and optimized (red dot) emissions are compared with ground-based observations from the WDCGG (a) and TCCON networks (b), respectively. The 2010–2017 trends that are significant in a statistical test ($p < 0.05$) observed by WDCGG (c) and TCCON (d) are used to evaluate the modelled trends. The trends are calculated based on monthly time series using a curve fitting method as described in Zheng et al. (2018a).