# Peer review of "Global atmospheric carbon monoxide budget 2000–2017 inferred from multi-species atmospheric inversions"

_Earth System Science Data, 2019_

## Referee Comment (RC1) · Anonymous Referee #1 · 29 May 2019

Zheng et al present an analysis of the global CO budget for 2000 – 2017, based on satellite retrievals of total column CO, CH4 and formaldehyde. The study uses a Bayesian inversion approach to optimize CO sources and the sink in a 3-D chemistry-transport model. A detailed analysis of CO budget trends by region is presented, allowing for some conclusions regarding the types of CO sources driving the trends. The authors also compare their results to bottom-up CO emission inventories, other inversion studies as well as surface-based observations.

Summary Statement

CO is an important trace gas, both because of its key role in global atmospheric chem-

istry and because it is a regulated pollutant; the study topic is thus of high interest. I am not an expert in satellite retrieval approaches, but the good overall agreement of the results with both bottom-up emission inventories and surface-based observations is convincing. The confirmation of the continuing decline of the atmospheric CO burden and CO emissions is a useful result that is well suited for publication in ESSD. The analysis is very thorough, in my opinion, unambiguously identifying a downward trend in anthropogenic CO emissions and also reinforcing earlier work that suggested that biomass burning emissions likely also declined overall.

Specific Comments:

Comparison with ground-based observations is an important part of validating the model. I would recommend moving one of the associated figures (e.g., S8) into the main paper to increase its visibility.

While the modeled results agree well with surface CO measurements for most stations (WDCGG comparison), there are many outliers as well. Some discussion of these disagreements (which are quite large for some stations) is needed.

Several of the figures and tables (Figs 6, 7bcef, 8, 9, 10, S11, Table S5) do not present uncertainty estimates. These should be added to the figures / tables. For some of the figures (e.g., Fig 8) perhaps just uncertainties for a subset of the traces (or just one of the traces, such as Inversion 1 in Fig 8) could be added to avoid overloading the figure.

Page 8, line 12. I think the value given here for the CO sink trend is wrong – it is inconsistent with what is described in Section 3.2.1 for Inversion 1; also if the sink is declining faster than the source, atmospheric CO would be going up, not down.

---

## Referee Comment (RC2) · Anonymous Referee #2 · 27 Jun 2019

This study present the atmospheric CO budget constrained by 4 different global inversions, including the discussion on the trends of emissions by sources and sectors. The paper is well written and provide a documented and interesting discussion of the results and should be published after revisions.

However, the authors stress results that are already known, which is the decline of the emissions in Europe and in the USA that has been going on for several decades, and from China since the 2010's, for which the trend is not well represented in bottom-up inventories. The inversion system has been designed for multi-species inversions (e.g. Pison et al. 2009), so it looks like that aside from updating MOPITT retrievals and the

model, the addition of GOSAT is the main update in the system. It would be great to have more discussion about the impact of GOSAT and the overall CH4 to CH2O to CO yield within that context. In particular, the direct comparison to CO observations does even suggest a slight deterioration when using GOSAT (inversion #3, Fig. S4 and S8). Then it is desirable to see the impact of CH2O and CH4 (inversion #2 and #3) on the spatial and temporal distribution of CO sources and sinks (showing Fig. 1 and Fig. 4 with inversion #2 and #3). It is surprising that the CO dry deposition is not presented, when the focus of the paper is about the CO budget, please take into account the following major and minor comments before resubmission.

Major comments: 1. Why is the dry deposition of CO not included in the Eq. 1? The dry deposition constitutes around 10 % (up to 15 %) of the total sink and it has been showed that an alternative representation of the CO deposition could improve the atmospheric CO budget (see Stein et al. 2014 and reference therein). If you did not take the dry deposition into account, it means that the global CO budget has an uncertainty of 10 to 15 % error bar, which likely introduce a bias such as underestimation of emissions or an overestimation of the chemical sink. This precludes strong conclusions about the temporal variations of the CO budget, in particular with regard to the chemical sink. Please correct the equation 1 and the following: P6L1; "The picture of atmospheric CO budget derived from our inversions includes surface emissions from different source sectors, CO chemical production, and CO chemical sink." P2L22: "Interpreting atmospheric CO trends requires accurate quantification of the global CO budget (Duncan et al., 2007), including surface sources, atmospheric sources (oxidation of hydrocarbons, known as CO chemical production), and atmospheric sinks."

2. The publication must discuss and includes reference to the limitation of using a simpler and linearized chemical scheme. This is one of the main points of the already cited paper; Lelieveld et al. (2016) showed that OH is well buffered because of the secondary OH production (OH recycling), which is not represented in your system. It has two implications, one is the absence of atmospheric chemical feedbacks given

their distribution of chemicals and the second is the impact of the ignored species. For instance, NOx emissions have a strong impact on OH (e.g. Miyazaki et al. 2017). Several studied have pointed out that because of the coupling between CH4, CO and OH, their errors are potentially corelated and thus reducing one bias could lead to an overall benefit for the whole system (e.g., Strode et al. 2015, Gaubert et al. 2016). In particular one recent study showed that the CH4 observations could thus be used to constraint OH feedback (Zhang et al. 2018). The use of a simplified chemistry is totally acceptable knowing the computational costs for studying long-term trends, as done in this study. It is however important to point out those potential limitation when points are made on the constraints on the CO (+OH) sinks. One easy thing to do would be to divide the CH4 loss by the CH4 concentrations used (i.e. the CH4 lifetime), since there is a, to check whether the growing CH4 sink is due to increasing CH4 alone or if there is a change due to a change in OH.

Minor comments:

P4L12: "To solve the inverse problem, forward and adjoint codes are iteratively run until sufficient convergence of the cost function (Eq. (2)), and the last iteration with optimized model states gives us the best estimate that matches all available information within their uncertainties." Please move this sentence to the next section, where Eq. 2 is actually defined.

P5L23: "The WDCGG measures surface hourly CO concentrations". It looks like there are flasks measurements, please rephrase.

P5L13: Add the references for GOSAT (Kuze et al. 2009) and GOSAT CH4 retrievals (Parker et al. 2011). [Kuze, A., Suto, H., Nakajima, M., and Hamazaki, T.: Thermal and near infrared sensor for carbon observation Fourier transform spectrometer on the Greenhouse Gases Observing Satellite for greenhouse gases monitoring, Appl. Opt., 48, 6716, https://doi.org/10.1364/ao.48.006716, 2009.]

P6L3: "which calculates the CO yield from the oxidation of CH4 and of NMVOCs and

the CO oxidation sink in each model grid box at each time step of the model simulation."
Here you can recall that the yield is assume to be linear.

P8L2: "Tropospheric CO columns measured by MOPITT have declined at a relative
rate of $-0.32\pm0.05\%$ yr$-1$ (p<0.01) during 2000–2017, highly consistent with the rel-
ative trend in the estimated CO sink ($-0.35\pm0.23\%$ yr$-1$, p<0.01). This suggests that
decreasing CO concentrations are the primary driver of the declining CO sink, while
the combination of OH and reaction rate has negligible influence." While I agree that
based on other evidence the CO emissions are reduced, which leads to a reduced CO
levels, and thus the CO sink is reduced. MOPITT also see a reduction in CO concen-
trations. But this does not mean that OH does not have an influence, understanding
OH feedbacks are far more complicated, since it requires the understanding of the OH
budget itself. One can only say that the CO + OH flux has slowed down because of a
reduction of CO. This also means that the OH sink has slowed down, but it does not
fully explain the OH budget. This is important because even a tiny change in OH would
have a strong impact on CO, so that it is hard to identify those small changes precisely.

P8L7: "These two inversions make a small difference (<10%) in the global CO budget
estimates compared to Inversion #1 (Table S5)" Why is there only one significant fig-
ure? In particular for differences within 10 %, it makes it impossible to compare the
simulations. This is also true for table S6.

P10L13: "The global CO sink is symmetrically distributed around the equator (Fig. 3a,
4c)." Is this due to the (TRANSCOM) prior?

P11L15: "As Inversion #1 tends to underestimate/overestimate anthropogenic emis-
sions decrease/increase (Sect. 3.2)," Is there something missing? You mean in corre-
sponding regions? it underestimates the increase in regions where CO is decreasing
and vice versa? Please rephrase.

P11L17: "the CEDS inventory probably has large biases in emission trends estimates
over CHN and SAS, which is the main reason why it estimates growing anthropogenic

emissions globally (Table S6) and cannot match the observed declining CO when used in the input of our LMDz-SACS model." You can note that this is consistent with previous studies, in particular Strode et al. 2016, for a different inventory.

Figure S4/S5/S6. In panel a, "Opimized"

P12L3: "The larger biomass burning emissions derived from inversions are most evident in late fire seasons when burned area declines after the peak fire month (Fig. S2)." It is also evident for the peak itself for SAF and BRA.

P14L18: "The global burned area is observed to have declined since 2000 (Fig. 9a) with the largest declines in the grassland and savanna ecosystems over EQAF (Fig. 9b)." I guess "decline" should not have an s

P16L18: "The other three inversions"; you can mention that those are also multi-species inversions, since it was mentioned for the previous paragraph.

References

Miyazaki, K., Eskes, H., Sudo, K., Boersma, K. F., Bowman, K., and Kanaya, Y.: Decadal changes in global surface NOx emissions from multi-constituent satellite data assimilation, Atmos. Chem. Phys., 17, 807-837, https://doi.org/10.5194/acp-17-807-2017, 2017.

Gaubert, B., Arellano, A. F., Barré, J., Worden, H. M., Emmons, L. K., Tilmes, S., Buchholz, R. R.,Vitt, F., Raeder, K., Collins, N., Anderson, J. L., Wiedinmyer, C., Martinez Alonso, S., Edwards, D. P., Andreae, M. O., Hannigan, J. W., Petri, C., Strong, K., and Jones, N.: Toward a chemical reanalysis in a coupled chemistry-climate model: An evaluation of MOPITT CO assimilation and its impact on tropospheric composition, J. Geophys. Res.-Atmos., 121, 7310–7343, https://doi.org/10.1002/2016JD024863, 2016.

Lelieveld, J., Gromov, S., Pozzer, A., and Taraborrelli, D.: Global tropospheric hydroxyl distribution, budget and reactivity, Atmos. Chem. Phys., 16, 12477-12493, doi:

10.5194/acp-16-12477-2016, 2016.

Stein, O., Schultz, M. G., Bouarar, I., Clark, H., Huijnen, V., Gaudel, A., George, M., and Clerbaux, C.: On the wintertime low bias of Northern Hemisphere carbon monoxide found in global model simulations, Atmos. Chem. Phys., 14, 9295-9316, https://doi.org/10.5194/acp-14-9295-2014, 2014.

Strode, S. A., Duncan, B. N., Yegorova, E. A., Kouatchou, J., Ziemke, J. R., and Douglass, A. R.: Implications of carbon monoxide bias for methane lifetime and atmospheric composition in chemistry climate models, Atmos. Chem. Phys., 15, 11789-11805, https://doi.org/10.5194/acp-15-11789-2015, 2015.

Strode, S. A., Worden, H. M., Damon, M., Douglass, A. R., Duncan, B. N., Emmons, L. K., Lamarque, J.-F., Manyin, M., Oman, L. D., Rodriguez, J. M., Strahan, S. E., and Tilmes, S.: Interpreting space-based trends in carbon monoxide with multiple models, Atmos. Chem. Phys., 16, 7285-7294, https://doi.org/10.5194/acp-16-7285-2016, 2016.

Zhang, Y., Jacob, D. J., Maasakkers, J. D., Sulprizio, M. P., Sheng, J.-X., Gautam, R., and Worden, J.: Monitoring global tropospheric OH concentrations using satellite observations of atmospheric methane, Atmos. Chem. Phys., 18, 15959-15973, https://doi.org/10.5194/acp-18-15959-2018, 2018.

---

## Referee Comment (RC3) · Anonymous Referee #3 · 28 Jun 2019

General Comments:

The authors provide a detailed analysis to constrain CO emissions with multi-satellite measurements in the period of 2000-2017. They demonstrated decreasing trends of anthropogenic and biomass burning emissions, and noticeable influences from the assimilation of HCHO on the estimation of oxidation sources. I found their paper is interesting and helpful for people in this field. I recommend the paper for publication after consideration of the points below.

Specific Comments:

1. Abstract: I am not sure whether the biased trends in the bottom-up inventories are

still "surprising", as the bias has been found with inverse analysis several years ago.

2. Section 3.2.2: Will INV #1 and INV #2/3 have better agreement in wintertime, when the contribution from NMVOCs is smaller?

3. Figure 2b: As the largest difference is in China, it will be helpful to check whether the a posteriori simulations of INV #2/#3 match better with surface measurements in China outflow regions than that of INV #1.

4. Page 9, Line 17-18: "Therefore, it is reasonable to think that Inversion #3 has a more realistic representation of the global CO budget than Inversion #2 does, and Inversion #2 is better than Inversion #1."

It may not be as obvious as mentioned here. I agree the observations of HCHO/CH4 will be helpful to distinguish the sources from combustion and oxidation, however, why they will improve the global CO budget? The assimilation of HCHO/CH4 will affect OH, but the ability of global models to simulate OH chemistry is still weak.

5. Figure 5b: the trends are generally positive in India and negative in the rest of SEA, which is surprising. I have assumed that they will be similar.

6. Page 12, Lines 29-32: The validation with independent surface measurements is an essential part in this work. These figures should be included in the main text rather than supplement.

I found the numbers for different periods are compared directly, which will affect the reliability of the validation: INV #1 Figure S4c, 2000-2017 INV #2 Figure S6c, 2005-2017 INV # 3 Figure S8c, 2010-2017

In addition, the distributions of data points are very noisy. I cannot see any noticeable difference among those figures by my eyes.

7. Page 14, Line 24: The author name in the citation.

---

## Author Comment (AC1) · 3 Aug 2019

*Reviewer #1:*

*Comments:*

*Zheng et al present an analysis of the global CO budget for 2000–2017, based on satellite retrievals of total column CO, CH4 and formaldehyde. The study uses a Bayesian inversion approach to optimize CO sources and the sink in a 3-D chemistry-transport model. A detailed analysis of CO budget trends by region is presented, allowing for some conclusions regarding the types of CO sources driving the trends. The authors also compare their results to bottom-up CO emission inventories, other inversion studies as well as surface-based observations.*

*Summary Statement*

*CO is an important trace gas, both because of its key role in global atmospheric chemistry and because it is a regulated pollutant; the study topic is thus of high interest. I am not an expert in satellite retrieval approaches, but the good overall agreement of the results with both bottom-up emission inventories and surface-based observations is convincing. The confirmation of the continuing decline of the atmospheric CO burden and CO emissions is a useful result that is well suited for publication in ESSD. The analysis is very thorough, in my opinion, unambiguously identifying a downward trend in anthropogenic CO emissions and also reinforcing earlier work that suggested that biomass burning emissions likely also declined overall.*

**Response:**

We would like to thank the referee for the positive comments on our manuscript. Below is our response to specific comments. The corresponding changes to the initial text are marked in red.

*Specific Comments:*

*Comparison with ground-based observations is an important part of validating the model. I would recommend moving one of the associated figures (e.g., S8) into the main paper to increase its visibility.*

**Response:**

We have moved Figs. S4, S6, and S8 into the main text to make it easier for readers to access them.

*While the modeled results agree well with surface CO measurements for most stations (WDCGG comparison), there are many outliers as well. Some discussion of these disagreements (which are quite large for some stations) is needed.*

**Response:**

We have added a discussion of the disagreements in Sect. 3.4 of the revised manuscript as follows.

"The WDCGG sites that show large disagreements are mostly located at coastal terrain areas, where our coarse-resolution model simplifies the coastline and thus cannot resolve the associated meteorology well (e.g., land-sea breeze circulation) (Palau et al., 2005; Ahmadov et al., 2007) and possible local emission sources. Several sites at high northern latitudes also suggest relatively large modelling bias due to the lack of high-quality satellite data as an observational constraint."

*Several of the figures and tables (Figs 6, 7bcef, 8, 9, 10, S11, Table S5) do not present uncertainty estimates. These should be added to the figures / tables. For some of the figures (e.g., Fig 8)*

*perhaps just uncertainties for a subset of the traces (or just one of the traces, such as Inversion 1 in Fig 8) could be added to avoid overloading the figure.*

**Response:**

We did not add uncertainty error bars in Figs 6, 8, 9, 10, S11, and Table S5 because a variational Bayesian inversion system cannot quantify the uncertainty of the inversion fluxes directly. The estimate of uncertainties would need a Monte Carlo method that is computationally expensive for studying long-term trends. Therefore we used sensitivity tests that varied prior fluxes and observational constraints to assess the uncertainty in this study.

The uncertainties of the trend estimate in Figs. 7bcef are provided in Tables S7 and S10. We did not present them in Fig. 7 because the area of each dot is proportional to annual average emissions, which makes it difficult to add the error bars in the same figure.

*Page 8, line 12. I think the value given here for the CO sink trend is wrong – it is inconsistent with what is described in Section 3.2.1 for Inversion 1; also if the sink is declining faster than the source, atmospheric CO would be going up, not down.*

**Response:**

The values given here are correct. The slightly faster-decreasing rate of the CO sink ($-11.3\pm11.0$ Tg CO yr$^{-2}$) than the CO source ($-10.3\pm12.7$ Tg CO yr$^{-2}$) during 2005–2017 would not make the atmospheric CO go up, because the atmospheric CO has kept declining since 2000, and the CO total sink is estimated to be larger than the CO total source during the whole period of 2005–2017.

**References**

Ahmadov, R., Gerbig, C., Kretschmer, R., Koerner, S., Neininger, B., Dolman, A. J., and Sarrat, C.: Mesoscale covariance of transport and $CO_2$ fluxes: Evidence from observations and simulations using the WRF-VPRM coupled atmosphere-biosphere model, J. Geophys. Res. Atmos., 112, doi: 10.1029/2007JD008552, 2007.

Palau, J. L., Pérez-Landa, G., Diéguez, J. J., Monter, C., and Millán, M. M.: The importance of meteorological scales to forecast air pollution scenarios on coastal complex terrain, Atmos. Chem. Phys., 5, 2771-2785, doi: 10.5194/acp-5-2771-2005, 2005.

---

## Author Comment (AC2) · 3 Aug 2019

*Reviewer #2:*

*Comments:*

*This study present the atmospheric CO budget constrained by 4 different global inversions, including the discussion on the trends of emissions by sources and sectors. The paper is well written and provide a documented and interesting discussion of the results and should be published after revisions.*

**Response:**

We thank the referee for his/her careful and constructive review. Our point-by-point responses to the reviewer's concerns are presented below. The changes to the initial text are marked in red.

*However, the authors stress results that are already known, which is the decline of the emissions in Europe and in the USA that has been going on for several decades, and from China since the 2010's, for which the trend is not well represented in bottom-up inventories.*

**Response:**

This is a data description paper that evaluates a new global CO budget product and makes the data publically available for further analysis. Several key points that are new compared to previous studies: 1) the longest time period covering 2000–2017; 2) the assessment of multi-species data assimilation; 3) the evaluation of biases in the anthropogenic emissions data CEDS, the latest global inventory that is used in the CMIP6; and 4) an evaluation of uncertainty via sensitivity tests.

*The inversion system has been designed for multi-species inversions (e.g. Pison et al. 2009), so it looks like that aside from updating MOPITT retrievals and the model, the addition of GOSAT is the main update in the system.*

**Response:**

There are several important updates compared to the initial system developed by Pison et al. (2009). We use a higher resolution transport model combined with the latest prior emission inventory, assimilate the new MOPITT retrievals, and also add OMI HCHO and GOSAT $XCH_4$ constraints.

*It would be great to have more discussion about the impact of GOSAT and the overall CH4 to $CH_2O$ to CO yield within that context. In particular, the direct comparison to CO observations does even suggest a slight deterioration when using GOSAT (inversion #3, Fig. S4 and S8).*

**Response:**

We have presented the impact of adding OMI HCHO and GOSAT $XCH_4$ constraints in Sect. 3.2.2, and have discussed the implication for the CO budget analysis in Sect. 3.2.3. The key finding is that the multi-species inversions have a more realistic representation of the source splitting between anthropogenic emissions and chemical production in the CO budget estimates.

The modelling performance of Inversion #3 (Fig. S8) is broadly comparable with that of Inversion #1 (Fig. S4) for annual averages of CO and XCO and it slightly improves the trend estimates. Overall the improvement of the posterior simulation is not significant, because the additional assimilation of $HCHO/CH_4$ increases the declining trend of anthropogenic emissions, but also increases the CO chemical production, which does not change the CO total source much.

*Then it is desirable to see the impact of CH$_2$O and CH$_4$ (inversion #2 and #3) on the spatial and temporal distribution of CO sources and sinks (showing Fig. 1 and Fig. 4 with inversion #2 and #3).*

**Response:**

We have created similar figures like Fig. 1 and Fig. 4 with the inversion results from #2 and #3 as the reviewer suggested. These new figures have been added in the supplement material.

*It is surprising that the CO dry deposition is not presented, when the focus of the paper is about the CO budget, please take into account the following major and minor comments before resubmission.*

*Major comments: 1. Why is the dry deposition of CO not included in the Eq. 1? The dry deposition constitutes around 10 % (up to 15 %) of the total sink and it has been showed that an alternative representation of the CO deposition could improve the atmospheric CO budget (see Stein et al. 2014 and reference therein). If you did not take the dry deposition into account, it means that the global CO budget has an uncertainty of 10 to 15% error bar, which likely introduce a bias such as underestimation of emissions or an overestimation of the chemical sink. This precludes strong conclusions about the temporal variations of the CO budget, in particular with regard to the chemical sink. Please correct the equation 1 and the following: P6L1; "The picture of atmospheric CO budget derived from our inversions includes surface emissions from different source sectors, CO chemical production, and CO chemical sink." P2L22: "Interpreting atmospheric CO trends requires accurate quantification of the global CO budget (Duncan et al., 2007), including surface sources, atmospheric sources (oxidation of hydrocarbons, known as CO chemical production), and atmospheric sinks."*

**Response:**

The dry deposition of CO is not directly represented by the LMDz-SACS model (Pison et al. 2009). As a consequence, the prior simulation suffers from a small bias (as illustrated by Stein et al. (2014) using other models) that adds to the other prior biases. However, the inversion system still optimizes the surface flux of CO, which is in practice the sum of CO surface emissions and CO dry deposition in the boundary layer. Using the reviewer's number (around 10 % of the total sink) combined with ours from Table 3, we get an average dry deposition budget about 20% of the net CO flux. We consider this to be marginal in view of the other uncertainty factors and have been neglecting it in the analysis of our results so far. We agree with the reviewer that this simplification should be clearly stated and we have made the following modifications.

First, we correct Equation 1 as:

$$\frac{\partial [CO]}{\partial t} = \sum (Source_{CO}) - Sink_{CO}$$

$$= -\mathbf{v} \bullet \nabla [CO] + \sum_{sector} (E_{CO}) + P_{CH_4 \to CO} + P_{NMVOCs \to CO} - k_{CO+OH}(T)[CO][OH] - Dep_{CO}$$

P4L5: "The CO chemical sink ($k_{CO+OH}(T)[CO][OH]$) is calculated on the basis of CO ($[CO]$), OH ($[OH]$), and a temperature ($T$)-dependent rate ($k_{CO+OH}$), and $Dep_{CO}$ is the dry deposition of CO that contributes about 7% of the CO total sink (Stein et al., 2014)."

P4L7: "We use the global 3-D transport model of the Laboratoire de Météorologie Dynamique (LMDz) coupled with a simplified chemistry module, Simplified Atmospheric Chemistry assimilation System (SACS) (Pison et al., 2009), to simulate the atmospheric physical and chemical processes described in Eq. (1) except the dry deposition not represented by this model."

P6L2: "The picture of atmospheric CO budget derived from our inversions includes surface fluxes (the sum of direct emissions from different source sectors and of dry deposition), CO chemical production, and CO chemical sink. Given the marginal role played by dry deposition (about 20% of the direct emissions, Stein et al. 2014), the inferred surface fluxes will be assumed to be made of direct emissions only in the following."

*2. The publication must discuss and includes reference to the limitation of using a simpler and linearized chemical scheme. This is one of the main points of the already cited paper; Lelieveld et al. (2016) showed that OH is well buffered because of the secondary OH production (OH recycling), which is not represented in your system. It has two implications, one is the absence of atmospheric chemical feedbacks given their distribution of chemicals and the second is the impact of the ignored species. For instance, NOx emissions have a strong impact on OH (e.g. Miyazaki et al. 2017). Several studied have pointed out that because of the coupling between CH₄, CO and OH, their errors are potentially correlated and thus reducing one bias could lead to an overall benefit for the whole system (e.g., Strode et al. 2015, Gaubert et al. 2016). In particular one recent study showed that the CH₄ observations could thus be used to constraint OH feedback (Zhang et al. 2018). The use of a simplified chemistry is totally acceptable knowing the computational costs for studying long-term trends, as done in this study. It is however important to point out those potential limitation when points are made on the constraints on the CO (+OH) sinks. One easy thing to do would be to divide the CH₄ loss by the CH₄ concentrations used (i.e. the CH₄ lifetime), since there is a, to check whether the growing CH₄ sink is due to increasing CH₄ alone or if there is a change due to a change in OH.*

**Response:**

We have added a new paragraph in Sect. 4.3 in the revised manuscript to address this point:

"The ability to simulate the nonlinear chemistry of OH is still weak in global models, which is another challenge to understand the OH variation. The LMDz-SACS model adopts a linearized chemical scheme to simulate the hydrocarbon reactions including CH₄+OH, HCHO+OH, CO+OH, and MCF+OH. The nonlinear dynamics of the OH chemistry, such as the secondary OH production (Lelieveld et al., 2016) and the interaction of OH with the NOₓ chemistry (Miyazaki et al., 2017), is not represented. The negligible computational cost of this configuration for OH motivates it, but we also expect the optimization of OH through the joint assimilation of CH₄, HCHO, CO, and MCF observations to counterbalance the simplicity of the scheme. Alternatively, it would be interesting to sophisticate the scheme by introducing key tracers in the OH chemistry (e.g., tropospheric ozone, NO, and NMVOCs) in the scheme together with prescribed (though uncertain) reaction rates, but we currently lack enough observations to constrain this additional complexity."

The inversion-based OH trend is presented in the supplement material, and is also compared with other studies and discussed with details in Sect. 4.3.

*Minor comments:*

*P4L12: "To solve the inverse problem, forward and adjoint codes are iteratively run until sufficient convergence of the cost function (Eq. (2)), and the last iteration with optimized model states gives us the best estimate that matches all available information within their uncertainties." Please move this sentence to the next section, where Eq. 2 is actually defined.*

**Response:**

This sentence has been moved to Section 2.2 after the Eq. 2 is defined.

*P5L23: "The WDCGG measures surface hourly CO concentrations". It looks like there are flasks measurements, please rephrase.*

**Response:**

This sentence has been rephrased as "The WDCGG provides measurements of surface CO concentrations through in-situ and flask sample measurements".

*P5L13: Add the references for GOSAT (Kuze et al. 2009) and GOSAT CH$_4$ retrievals (Parker et al. 2011). [Kuze, A., Suto, H., Nakajima, M., and Hamazaki, T.: Thermal and near infrared sensor for carbon observation Fourier transform spectrometer on the Greenhouse Gases Observing Satellite for greenhouse gases monitoring, Appl. Opt., 48, 6716, https://doi.org/10.1364/ao.48.006716, 2009.]*

**Response:**

The references for GOSAT (Kuze et al. 2009) and GOSAT CH$_4$ retrievals (Parker et al. 2011) have been added in Table 1 that describes configurations of the inversion system.

*P6L3: "which calculates the CO yield from the oxidation of CH$_4$ and of NMVOCs and the CO oxidation sink in each model grid box at each time step of the model simulation." Here you can recall that the yield is assume to be linear.*

**Response:**

As suggested by the reviewer, this sentence is changed to "which calculates the CO yield from the oxidation of CH$_4$ and of NMVOCs and the CO oxidation sink with the linearized chemistry scheme in each model grid box at each time step of the model simulation".

*P8L2: "Tropospheric CO columns measured by MOPITT have declined at a relative rate of −0.32±0.05% yr$^{-1}$ (p<0.01) during 2000–2017, highly consistent with the relative trend in the estimated CO sink (−0.35±0.23% yr$^{-1}$, p<0.01). This suggests that decreasing CO concentrations are the primary driver of the declining CO sink, while the combination of OH and reaction rate has negligible influence." While I agree that based on other evidence the CO emissions are reduced, which leads to a reduced CO levels, and thus the CO sink is reduced. MOPITT also see a reduction in CO concentrations. But this does not mean that OH does not have an influence, understanding OH feedbacks are far more complicated, since it requires the understanding of the OH budget itself. One can only say that the CO + OH flux has slowed down because of a reduction of CO. This also means that the OH sink has slowed down, but it does not fully explain the OH budget. This is important because even a tiny change in OH would have a strong impact on CO, so that it is hard to identify those small changes precisely.*

**Response:**

We have rewritten this sentence in this way: "This suggests that decreasing CO concentrations are the primary driver of the declining CO sink, and dominate over the influence from the possible changes in OH and reaction rate.".

*P8L7: "These two inversions make a small difference (<10%) in the global CO budget estimates compared to Inversion #1 (Table S5)" Why is there only one significant figure? In particular for differences within 10 %, it makes it impossible to compare the simulations. This is also true for table S6.*

**Response:**

The value of 10% represents the largest difference for a single year. Table S5 and S6 both present the multiannual averages that only have less than 2% difference between different inversions. That's why it is difficult to compare the difference in these two tables. We have clarified this in the revised manuscript.

"These two inversions make a small difference (<10% for a single year and <2% for multiannual mean) in the global CO budget estimates compared to Inversion #1 (Table S5)"

*P10L13: "The global CO sink is symmetrically distributed around the equator (Fig. 3a, 4c)." Is this due to the (TRANSCOM) prior?*

**Response:**

Our inversion results actually suggest that the CO chemical sink is 30% larger in the Northern Hemisphere than that in the Southern Hemisphere mainly due to the higher CO levels in the Northern Hemisphere.

We have rewritten this sentence as follows: "The global CO sink presents an asymmetrical distribution around the equator that is 30% larger in the Northern Hemisphere than that in the Southern Hemisphere, due to the higher CO levels in the Northern Hemisphere".

*P11L15: "As Inversion #1 tends to underestimate/overestimate anthropogenic emissions decrease/increase (Sect. 3.2)," Is there something missing? You mean in corresponding regions? it underestimates the increase in regions where CO is decreasing and vice versa? Please rephrase.*

**Response:**

We have rephrased this sentence as "As Inversion #1 tends to underestimate the decrease and to overestimate the increase of anthropogenic emissions (Sect. 3.2)".

*P11L17: "the CEDS inventory probably has large biases in emission trends estimates over CHN and SAS, which is the main reason why it estimates growing anthropogenic emissions globally (Table S6) and cannot match the observed declining CO when used in the input of our LMDz-SACS model." You can note that this is consistent with previous studies, in particular Strode et al. 2016, for a different inventory.*

**Response:**

We have added a sentence at the end of this paragraph: "This is consistent with the finding of Strode et al. (2016) who performed global CO modelling with a different model and inventory.".

*Figure S4/S5/S6. In panel a, "Opimized"*

**Response:**

Corrected.

*P12L3: "The larger biomass burning emissions derived from inversions are most evident in late fire seasons when burned area declines after the peak fire month (Fig. S2)." It is also evident for the peak itself for SAF and BRA.*

**Response:**

Yes. This sentence has been rephrased as "The larger biomass burning emissions derived from inversions are most evident in the peak fire month and in late fire seasons when burned area declines after the peak fire month (Fig. S2).".

*P14L18: "The global burned area is observed to have declined since 2000 (Fig. 9a) with the largest declines in the grassland and savanna ecosystems over EQAF (Fig. 9b)." I guess "decline" should not have an s*

**Response:**

Corrected.

*P16L18: "The other three inversions"; you can mention that those are also multispecies inversions, since it was mentioned for the previous paragraph.*

**Response:**

We have rephrased this sentence as "The other three inversions in the comparison are all derived from previous versions of our inversion system with multi-species constraints".

*References*

*Miyazaki, K., Eskes, H., Sudo, K., Boersma, K. F., Bowman, K., and Kanaya, Y.: Decadal changes in global surface NOx emissions from multi-constituent satellite data assimilation, Atmos. Chem. Phys., 17, 807-837, https://doi.org/10.5194/acp-17-807-2017, 2017.*

*Gaubert, B., Arellano, A. F., Barré, J., Worden, H. M., Emmons, L. K., Tilmes, S., Buchholz, R. R.,Vitt, F., Raeder, K., Collins, N., Anderson, J. L., Wiedinmyer, C., Martinez Alonso, S., Edwards, D. P., Andreae, M. O., Hannigan, J. W., Petri, C., Strong, K., and Jones, N.: Toward a chemical reanalysis in a coupled chemistry-climate model: An evaluation of MOPITT CO assimilation and its impact on tropospheric composition, J. Geophys. Res.-Atmos., 121, 7310–7343, https://doi.org/10.1002/2016JD024863, 2016.*

*Lelieveld, J., Gromov, S., Pozzer, A., and Taraborrelli, D.: Global tropospheric hydroxyl distribution, budget and reactivity, Atmos. Chem. Phys., 16, 12477-12493, doi: 10.5194/acp-16-12477-2016, 2016.*

*Stein, O., Schultz, M. G., Bouarar, I., Clark, H., Huijnen, V., Gaudel, A., George, M., and Clerbaux, C.: On the wintertime low bias of Northern Hemisphere carbon monoxide found in global model simulations, Atmos. Chem. Phys., 14, 9295-9316, https://doi.org/10.5194/acp-14-9295-2014, 2014.*

Strode, S. A., Duncan, B. N., Yegorova, E. A., Kouatchou, J., Ziemke, J. R., and Douglass, A. R.: Implications of carbon monoxide bias for methane lifetime and atmospheric composition in chemistry climate models, Atmos. Chem. Phys., 15, 11789-11805, https://doi.org/10.5194/acp-15-11789-2015, 2015.

Strode, S. A., Worden, H. M., Damon, M., Douglass, A. R., Duncan, B. N., Emmons, L. K., Lamarque, J.-F., Manyin, M., Oman, L. D., Rodriguez, J. M., Strahan, S. E., and Tilmes, S.: Interpreting space-based trends in carbon monoxide with multiple models, Atmos. Chem. Phys., 16, 7285-7294, https://doi.org/10.5194/acp-16-7285-2016, 2016.

Zhang, Y., Jacob, D. J., Maasakkers, J. D., Sulprizio, M. P., Sheng, J.-X., Gautam, R., and Worden, J.: Monitoring global tropospheric OH concentrations using satellite observations of atmospheric methane, Atmos. Chem. Phys., 18, 15959-15973, https://doi.org/10.5194/acp-18-15959-2018, 2018.

---

## Author Comment (AC3) · 3 Aug 2019

*Reviewer #3:*

*Comments:*

*General Comments:*

*The authors provide a detailed analysis to constrain CO emissions with multi-satellite measurements in the period of 2000-2017. They demonstrated decreasing trends of anthropogenic and biomass burning emissions, and noticeable influences from the assimilation of HCHO on the estimation of oxidation sources. I found their paper is interesting and helpful for people in this field. I recommend the paper for publication after consideration of the points below.*

**Response:**

We thank the referee for the positive and insightful comments. We have made a point-by-point response here. We have also marked the changes to the initial text in red.

*Specific Comments:*

*1. Abstract: I am not sure whether the biased trends in the bottom-up inventories are still "surprising", as the bias has been found with inverse analysis several years ago.*

**Response:**

We have removed the word "surprisingly" from the abstract.

*2. Section 3.2.2: Will INV #1 and INV #2/3 have better agreement in wintertime, when the contribution from NMVOCs is smaller?*

**Response:**

Yes. For example, Inv #1 and Inv #2 estimate the declining rates of $-1.6\%$ yr$^{-1}$ and $-2.0\%$ yr$^{-1}$, respectively, in annual anthropogenic emissions from China during 2005–2017. However, in wintertime (Jan–Mar) they estimate consistent trends of $-1.1\%$ yr$^{-1}$ and $-1.0\%$ yr$^{-1}$, respectively.

*3. Figure 2b: As the largest difference is in China, it will be helpful to check whether the a posteriori simulations of INV #2/#3 match better with surface measurements in China outflow regions than that of INV #1.*

**Response:**

We evaluated the a posteriori simulations with surface measurements over China and its outflow regions for Inv #1 (Figs. S5c, S5f), Inv #2 (Figs. S7c, S7f), and Inv #3 (Figs. S9c, S9f), respectively. These posterior simulations all correct modelling biases in the prior simulations, but Inv #2 and Inv #3 do not show significantly better performance than Inv #1. This is because the additional assimilation of HCHO/CH$_4$ increases the declining trend of anthropogenic emissions in China, but also increases the CO chemical production, which does not change the CO total source much.

*4. Page 9, Line 17-18: "Therefore, it is reasonable to think that Inversion #3 has a more realistic representation of the global CO budget than Inversion #2 does, and Inversion #2 is better than Inversion #1."*

*It may not be as obvious as mentioned here. I agree the observations of HCHO/CH4 will be helpful to distinguish the sources from combustion and oxidation, however, why they will improve the*

*global CO budget? The assimilation of HCHO/CH4 will affect OH, but the ability of global models to simulate OH chemistry is still weak.*

**Response:**

The reviewer's point is exactly what we discussed here. The previous sentence (Page 9, Line 16–17) in this paragraph said that "Constraining the CO chemical production can correct the inversion system that may inaccurately attribute some of the decreases in the CO source to the CO chemical production". Just to clarify, we have rephrased the sentence that the reviewer is concerned with as follows.

"Therefore, it is reasonable to think that Inversion #3 has a more realistic representation of the source splitting between anthropogenic emissions and chemical production in the global CO budget than Inversion #2 does, and Inversion #2 is better than Inversion #1."

*5. Figure 5b: the trends are generally positive in India and negative in the rest of SEA, which is surprising. I have assumed that they will be similar.*

**Response:**

The trends of anthropogenic emissions are estimated to have been growing in Indonesia but declining over most of mainland Southeast Asia, broadly consistent with MOPITT CO trends (Fig. 1a). However, the drivers behind are not clear yet due to lack of regional bottom-up inventories.

*6. Page 12, Lines 29-32: The validation with independent surface measurements is an essential part in this work. These figures should be included in the main text rather than supplement.*

*I found the numbers for different periods are compared directly, which will affect the reliability of the validation: INV #1 Figure S4c, 2000-2017 INV #2 Figure S6c, 2005-2017 INV # 3 Figure S8c, 2010-2017*

*In addition, the distributions of data points are very noisy. I cannot see any noticeable difference among those figures by my eyes.*

**Response:**

We have moved Figs. S4, S6, and S8 into the main text to make it easier for readers to access them.

We have rewritten Lines 29-32 Page 12 as follows to validate Inv #1, #2, and #3 at the same period.

"The evaluation with measurement from WDCGG suggests that Inversion #3 gives a fair estimate of surface CO trends during 2010–2017 (NMB = −8%, RMSE = 1.4 % $yr^{-1}$, Fig. B3c), while Inversion #2 (NMB = −34%, RMSE = 2.0 % $yr^{-1}$, Fig. B2c) and Inversion #1 (NMB = −47%, RMSE = 1.8 % $yr^{-1}$, Fig. B1c) still present moderate biases in their study period. During the overlap period of 2010–2017 with Inversion #3, Inversion #2 and Inversion #1 both present slightly larger RMSE of 1.5 % $yr^{-1}$ in the trend estimates."

The difference between these figures is marginal. Please refer to our response to the 3rd comment.

*7. Page 14, Line 24: The author name in the citation.*

**Response:**

This is not a citation but a reference to the year 2015. We have clarified the text.